# Dual separable feedback systems govern firing rate homeostasis

Yelena Kulik[1†], Ryan Jones[1†], Armen J Moughamian[2], Jenna Whippen[1], Graeme W Davis[1]*

[1]Department of Biochemistry and Biophysics, Kavli Institute for Fundamental Neuroscience, University of California, San Francisco, San Francisco, United States; [2]Department of Neurology, University of California, San Francisco, San Francisco, United States

**Abstract** Firing rate homeostasis (FRH) stabilizes neural activity. A pervasive and intuitive theory argues that a single variable, calcium, is detected and stabilized through regulatory feedback. A prediction is that ion channel gene mutations with equivalent effects on neuronal excitability should invoke the same homeostatic response. In agreement, we demonstrate robust FRH following either elimination of Kv4/Shal protein or elimination of the Kv4/Shal conductance. However, the underlying homeostatic signaling mechanisms are distinct. Eliminating Shal protein invokes *Krüppel*-dependent rebalancing of ion channel gene expression including enhanced *slo*, *Shab*, and *Shaker*. By contrast, expression of these genes remains unchanged in animals harboring a CRISPR-engineered, *Shal* pore-blocking mutation where compensation is achieved by enhanced $IK_{DR}$. These different homeostatic processes have distinct effects on homeostatic synaptic plasticity and animal behavior. We propose that FRH includes mechanisms of proteostatic feedback that act in parallel with activity-driven feedback, with implications for the pathophysiology of human channelopathies.
DOI: https://doi.org/10.7554/eLife.45717.001

*For correspondence:
graeme.davis@ucsf.edu

†These authors contributed equally to this work

## Introduction

Firing Rate Homeostasis (FRH) is a form of homeostatic control that stabilizes spike rate and information coding when neurons are confronted by pharmacological, genetic or environmental perturbation (*Davis, 2013*; *O'Leary et al., 2014*). FRH has been widely documented within invertebrate neurons (*Turrigiano et al., 1994*; *Muraro et al., 2008*; *Driscoll et al., 2013*) and neural circuits (*Haedo and Golowasch, 2006*) as well as the vertebrate spinal cord (*Gonzalez-Islas et al., 2010*), cortical pyramidal neurons (*Andrásfalvy et al., 2008*) and cardiomyocytes (*Guo et al., 2005*; *Marrus and Nerbonne, 2008*; *Michael et al., 2009*). In many of these examples, the genetic deletion of an ion channel is used to induce a homeostatic response. The mechanisms of FRH correct for the loss of the ion channel and precisely restore neuronal firing properties to normal, wild-type levels (*Swensen and Bean, 2005*; *Muraro et al., 2008*; *Andrásfalvy et al., 2008*; *Nerbonne et al., 2008*; *Van Wart and Matthews, 2006*; *Bergquist et al., 2010*; *Parrish et al., 2014*); see also: *MacLean et al., 2003*; *Ping and Tsunoda, 2012*; *Driscoll et al., 2013*). To date, little is understood about the underlying molecular mechanisms (but see *Parrish et al., 2014*; *Joseph and Turrigiano, 2017*; *Goold and Nicoll, 2010*; *Mee et al., 2004*).

FRH induced by an ion channel gene deletion is truly remarkable. The corrective response is not limited to the de novo expression of an ion channel gene with properties that are identical to the deleted channel, as might be expected for more generalized forms genetic compensation (*El-Brolosy and Stainier, 2017*; see also discussion). Instead, the existing repertoire of channels expressed by a neuron can be 'rebalanced' to correct for the deletion of an ion channel (*Swensen and Bean, 2005*; *Muraro et al., 2008*; *Andrásfalvy et al., 2008*; *Nerbonne et al., 2008*;

*Van Wart and Matthews, 2006*; *Bergquist et al., 2010*; *Parrish et al., 2014*; *Driscoll et al., 2013*). How is it possible to precisely correct for the absence of an essential voltage-gated ion channel? The complexity of the problem seems immense given that many channel types functionally cooperate to achieve the cell-type-specific voltage trajectory of an action potential.

Theoretical work argues that different mixtures of ion channels can achieve similar firing properties in a neuron (*Marder and Prinz, 2002*; *Marder and Goaillard, 2006*; *O'Leary et al., 2014*; *Golowasch, 2014*). These observations have led to a pervasive and intuitively attractive theory that a single physiological variable, calcium, is detected and stabilized through regulatory feedback control of ion channel gene expression (*O'Leary et al., 2014*). Yet, many questions remain unanswered. There are powerful cell biological constraints on ion channel transcription, translation, trafficking and localization in vivo (*Andrásfalvy et al., 2008*; *Carrasquillo and Nerbonne, 2014*). How do these constraints impact the expression of FRH? Is calcium the only intracellular variable that is sensed and controlled by homeostatic feedback? There remain few direct tests of this hypothesis (*Joseph and Turrigiano, 2017*). Why are homeostatic signaling systems seemingly unable to counteract disease-relevant ion channel mutations, including those that have been linked to risk for diseases such as epilepsy and autism (*Ben-Shalom et al., 2017*; *Klassen et al., 2011*)?

Here, we take advantage of the molecular and genetic power of *Drosophila* to explore FRH in a single, genetically identified neuron subtype. Specifically, we compare two different conditions that each eliminate the Shal/Kv4 ion channel conductance and, therefore, are expected to have identical effects on neuronal excitability. We demonstrate robust FRH following elimination of the Shal protein and, independently, by eliminating the Shal conductance using a pore blocking mutation that is knocked-in to the endogenous *Shal* locus. Thus, consistent with current theory, FRH can be induced by molecularly distinct perturbations to a single ion channel gene. However, we find that these two different perturbations induce different homeostatic responses, arguing for perturbation-specific effects downstream of a single ion channel gene.

Taken together, our data contribute to a revised understanding of FRH in several ways. First, altered activity cannot be *the sole* determinant of FRH. Two functionally identical manipulations that eliminate the Shal conductance, each predicted to have identical effects on neuronal excitability, lead to molecularly distinct homeostatic responses. Second, homeostatic signaling systems are sensitive to the type of mutation that affects an ion channel gene. This could have implications for understanding why FRH appears to fail in the context of human disease caused by ion channel mutations, including epilepsy, migraine, autism and ataxia. Finally, our data speak to experimental and theoretical studies arguing that the entire repertoire of ion channels encoded in the genome is accessible to the mechanisms of homeostatic feedback, with a very large combinatorial solution space (*Marder and Prinz, 2002*; *O'Leary et al., 2014*). Our data are consistent with the existence of separable proteostatic and activity-dependent homeostatic signaling systems, potentially acting in concert to achieve cell-type-specific and perturbation-specific FRH.

## Results

We first established a system to assess firing rate homeostasis following the elimination of the somatic A-type potassium channel encoded by the *Shal* gene, which contributes to the A-type potassium current ($IK_A$). To do so, we took advantage of the *GAL4-UAS* expression system for gene specific knockdown in *Drosophila melanogaster*. The *GAL4* line *MN1-GAL4* (previously referred to as *MN1-Ib-GAL4*; *Kim et al., 2009*) expresses selectively in a pair of segmentally repeated motoneurons that form synapses onto muscle 1 of the dorsal body wall (*Figure 1A*). We combined *MN1-GAL4* with a previously described *UAS-Shal-RNAi* that was shown to completely eliminate Shal protein when driven pan-neuronally (*Parrish et al., 2014*). Consistent with the previously documented effectiveness of the *Shal-RNAi* transgene, we found a dramatic reduction in somatically measured $IK_A$ when *Shal-RNAi* was driven by *MN1-GAL4* (*Figure 1B*). In wild-type MN1, $IK_A$ activated at approximately $-30$ mV and reached an average current density of 20 pA/pF at +40 mV. By contrast, no substantial current was present in MN1 expressing *Shal-RNAi* until +20 mV, and voltage steps above +20 mV revealed only a small outward current with $IK_A$ characteristics. Importantly, prior characterization of a *Shal* protein null mutation demonstrated the same current-voltage trajectory, including the same observed +50 mV shift in voltage activation (*Bergquist et al., 2010*). In that prior study, the remaining, voltage-shifted, outward current was determined to reflect the homeostatic

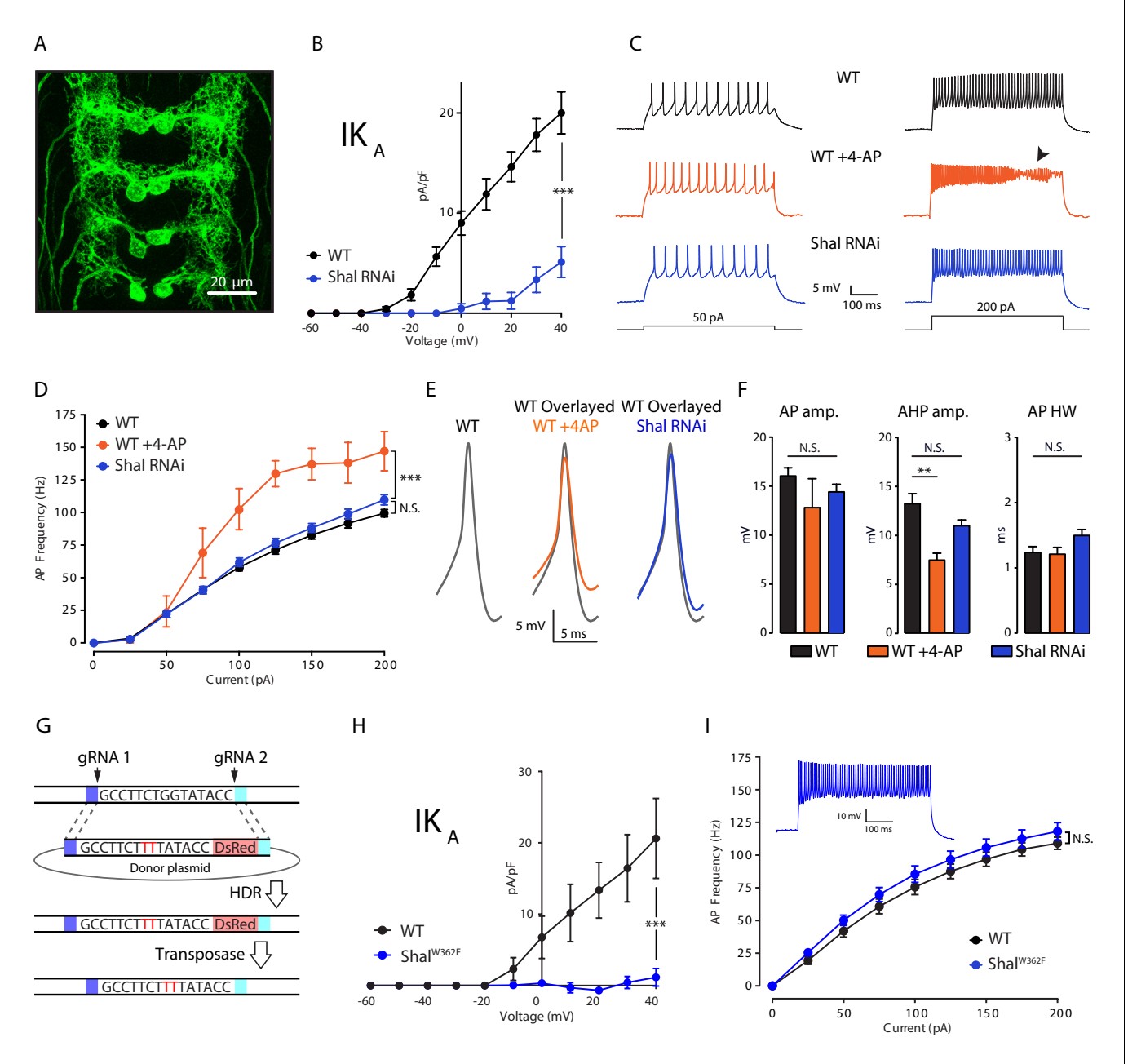

**Figure 1.** Firing rate homeostasis in *Drosophila* motoneurons. (A) Confocal max projection of *Drosophila* 3rd instar larval VNC shows selective Gal4 expression in MN1 (*MN1-GAL4 > UAS-CD8:GFP*). (B) I-V plots of MN1 $IK_A$ in WT (black, n = 20) and *Shal-RNAi* (blue, n = 10). (C) Representative voltage traces from WT (black), WT + 4 AP (orange) and *Shal-RNAi* (blue) at 50 pA (left) and 200 pA (right) current injections. Arrow indicates impaired action potentials due to depolarization block. (D) F-I curves of WT (n = 10), WT + 4 AP (n = 4) and *Shal-RNAi* (n = 15). (E) Example action potential waveforms (left) and overlay (right). (F) Quantification of action potential, after-hyperpolarization amplitudes (AP amp. AP AHP amp., respectively) and action potential half-width (AP HW). (G) CRISPR strategy for generating a targeted pore-blocking point mutation in *Shal*. Dark and light blue regions represent 5' and 3' 1 kb homology arms designed for recombination of mutated segment from pHD-ScarlessDsRed donor vector into the endogenous *Shal* gene locus. Selection marker represented in pink. (H) Elimination of $IK_A$ across all motoneurons in the *Shal*[W362F] mutant. WT (black, n = 7) and *Shal*[W362F] (blue, n = 13). (I) F-I curves of WT (n = 19) and *Shal*[W362F] (n = 15). Inset: Representative voltage trace from *Shal*[W362F] motoneuron at 200 pA current injection. Mean ± S.E.M.; *p<0.05; **p<0.005; ***p<0.0005. N.S., not significant; two-way RM-ANOVA with post-hoc tests (I-V plots and F-I curves) or one-way ANOVA with Bonferoni post-hoc tests (AP waveform measurements).

DOI: https://doi.org/10.7554/eLife.45717.002

The following figure supplements are available for figure 1:

**Figure supplement 1.** Action potential waveform measurements.

*Figure 1 continued on next page*

*Figure 1 continued*

DOI: https://doi.org/10.7554/eLife.45717.003

**Figure supplement 2.** 4-AP Does Not Increase Firing Rates in Shal$^{W362F}$ Motoneurons.

DOI: https://doi.org/10.7554/eLife.45717.004

upregulation of the Shaker channel, which resides in the electrotonically distant axonal membranes. This conclusion was independently confirmed in an additional, prior study (*Parrish et al., 2014*). Given these data, we conclude that *Shal-RNAi* effectively eliminated the relevant somatic IK$_A$ that would participate in action potential repolarization.

## Cell autonomous induction of firing rate homeostasis

To test for the cell autonomous induction of FRH, we compared the effects of acute pharmacological block of IK$_A$ using 4-aminopyridine (4-AP, 2.5 mM) with chronic IK$_A$ knockdown caused by *Shal-RNAi* expressed specifically in MN1 (*Figure 1C–F*). We demonstrate that application of 4-AP caused a significant increase in MN1 firing rate compared to wild type across all current steps greater than 50 pA (*Figure 1C & D*; WT + 4 AP, orange traces). At current steps above 150 pA, depolarization block was routinely observed, limiting the maximal firing rate that could be quantified (*Figure 1C*, arrowhead). Depolarization block was never observed in wild type. By contrast, Shal knockdown in MN1, eliminating somatic IK$_A$ (*Figure 1B*), did not result in a significant change in firing rate. Furthermore, depolarization block was never observed, just as in wild type (*Figure 1C & D*). Notably, 4-AP does not alter firing rate when applied to motoneurons lacking the Shal conductance, demonstrating the specificity of 4-AP at the concentration used in these experiments (see below). The differential effect of acute 4-AP versus chronic Shal knockdown can be taken as evidence for homeostatic, compensatory signaling that we define, here, as FRH. Based on this argument, we provide evidence that FRH can be induced and expressed in a single neuron.

## Homeostatic preservation of action potential waveform

To further investigate the precision of FRH, we examined the effects of 4-AP and Shal knockdown on action potential waveforms. Acute application of 4-AP caused a significant reduction in the afterhyperpolarization amplitude with no significant effect on amplitude or half-width (AHP, AP amp. and AP HW, respectively; *Figure 1E & F*; see *Figure 1—figure supplement 1*) for how these measurements are made). By contrast, no significant changes were observed when somatic IK$_A$ was eliminated selectively in MN1. We note that while 4-AP is a well-described IK$_A$ channel blocker (*Fedulova, 1999*; *Rudy, 1988*), it lacks complete specificity (*Kirsch and Drewe, 1993*). We can rule out a major contribution of Shaker to the 4-AP effect because Shaker channels are localized at an electrotonically distant site in the axon and presynaptic terminal (*Ford and Davis, 2014*; *Figure 1B*). Furthermore, the half-maximal effect of 4-AP on IK$_A$ in other systems (1–2 mM) is considered to have reasonable specificity and this concentration of 4-AP has quantitatively similar effects in *Drosophila* (*Ford and Davis, 2014*; *Jackson and Bean, 2007*). Regardless, it is remarkable that action potential repolarization and neuronal firing rate are statistically identical to wild type following the elimination of the Shal-mediated somato-dendritic IK$_A$ current. Thus, we demonstrate conservation of action potential waveform despite the absence of a primary fast potassium channel conductance (IK$_A$).

## Firing rate homeostasis induced by persistent elimination of the Shal conductance

We next asked whether FRH is induced when the Shal conductance is eliminated by a pore-blocking mutation. We used 'scarless' CRISPR-Cas9 gene editing technology (*Figure 1G*; *Gratz et al., 2015*) to engineer a point mutation in the *Shal* locus that renders the Shal channel non-conducting. This point mutation is a single amino acid substitution in the channel pore (W362F), a highly conserved mutation demonstrated to function as a pore-blocking mutation in systems as diverse as mammalian heterologous cells and cultured *Drosophila* embryonic neurons (*Barry et al., 1998*; *Ping et al., 2011*). We note that, both in vitro and in vivo, overexpression of this pore-blocked channel traffics to the plasma membrane (*Barry et al., 1998*; *Ping et al., 2011*). Here, we demonstrate that

motoneurons in the homozygous $Shal^{W362F}$ mutant lack somatically recorded $IK_A$ (*Figure 1H*). Next, we demonstrate the existence of robust FRH in $Shal^{W362F}$, and it is just as precise as that observed when Shal was eliminated using *UAS-Shal-RNAi* expressed in MN1 (*Figure 1I*). In contrast to wild-type motoneurons, bath application of 4-AP did not increase firing rates in $Shal^{W362F}$ mutant neurons (*Figure 1—figure supplement 2*). Thus, FRH can be induced by the loss of Shal channel function as well as loss of Shal protein.

## *Shal* knockdown induces FRH achieved by compensatory changes in $IK_{Ca}$ and IKDR

We hypothesized, based on work in *Drosophila* and other systems, that FRH is achieved by compensatory changes in ion channel gene expression (*Marder and Prinz, 2002*; *Nerbonne et al., 2008*; *Parrish et al., 2014*). Therefore, we assessed ionic conductances predicted to have a major role in controlling firing rate and action potential waveform including: the fast activating and inactivating potassium current $IK_A$, the delayed rectifier potassium current ($IK_{DR}$), the calcium-activated potassium current ($IK_{Ca}$), the voltage-gated sodium current ($I_{Na}$) and the voltage-gated calcium current ($I_{Ca}$). We found a significant enhancement of $IK_{DR}$ and $IK_{Ca}$, but no change in sodium or calcium currents in MN1 lacking Shal compared to wild type (*Figure 2A,B,C & D*, respectively). It was challenging to properly assess the total fast somatic sodium currents using standard voltage step protocols typically used in dissociated cells due to the inability to adequately maintain voltage control of the

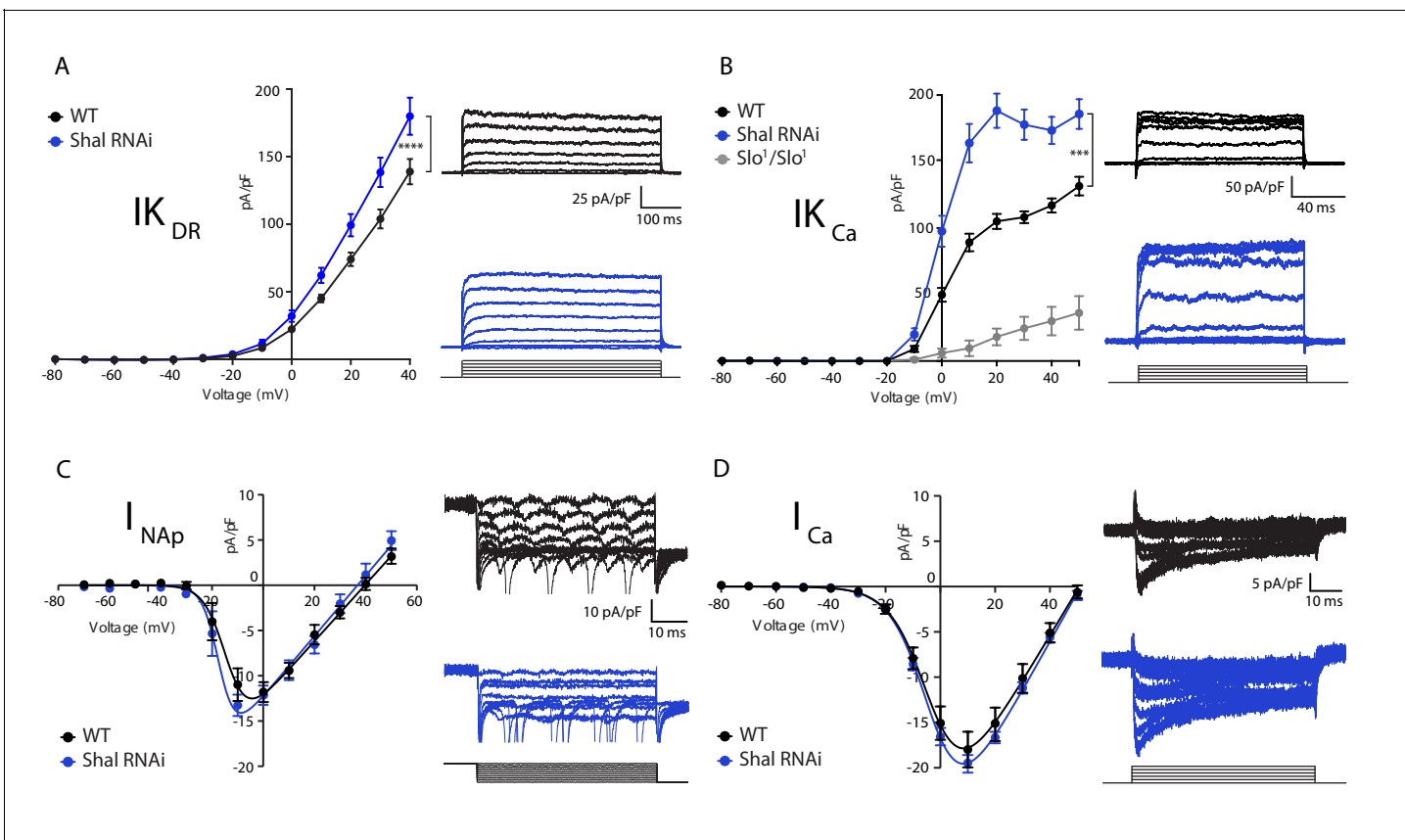

**Figure 2.** Homeostatic potassium current rebalancing stabilizes MN1 excitability in the absence of Shal-dependent $IK_A$. (A, B) I-V plots and representative traces of voltage-dependent delayed rectifier ($IK_{DR}$, (A) and $Ca^{2+}$-dependent ($IK_{Ca}$, (B) potassium currents in WT (black; n = 7 and n = 9 for $IK_{DR}$ and $IK_{Ca}$, respectively) and *Shal-RNAi* (blue; n = 12 and n = 8 for $K_{DR}$ and $K_{Ca}$, respectively) MN1. The specificity of the $IK_{Ca}$ current protocol was confirmed in $slo^1$ mutants (B, grey; n = 4), which exhibited minimal $Ca^{2+}$-dependent potassium currents. (C, D) I-V plots and representative traces of persistent sodium currents ($I_{NaP}$, C, WT: n = 9; *Shal-RNAi*: n = 9) and voltage dependent calcium currents ($I_{Ca}$, D; WT, n = 8; *Shal-RNAi*, n = 6) in WT (black) and *Shal-RNAi* (blue) MN1. Mean ± S.E.M.; *p<0.05; **p<0.005; ***p<0.0005. N.S., not significant; two-way RM-ANOVA with Bonferoni post-hoc tests (I-V plots).

DOI: https://doi.org/10.7554/eLife.45717.005

axon initial segment where action potentials are initiated. Therefore, we utilized a voltage-step protocol designed to isolate persistent sodium currents as a proxy for the total sodium current density (*French et al., 1990*; *Lin et al., 2009*). Although sodium spikes occasionally escaped voltage clamp (*Figure 2C*, traces), we were able to accurately measure persistent sodium current in both wild type and *Shal-RNAi* and found no significant change compared to wild-type MN1.

The largest compensatory conductance change that we observed following Shal knockdown was the enhancement of $IK_{Ca}$. To verify that our measurements were specific to calcium-dependent potassium currents, we performed the same protocol in the *slowpoke* (*slo*) mutant background, which eliminates the *Drosophila* BK channel ortholog (*Butler et al., 1993*; *Elkins et al., 1986*; *Komatsu et al., 1990*; *Singh and Wu, 1989*). The $IK_{Ca}$ current was virtually eliminated in the *slo* mutant (*Figure 2B*, gray line). We previously demonstrated that both BK and SK channel transcripts are increased in the *Shal*$^{495}$ null mutant background (*Parrish et al., 2014*). While we cannot rule out a contribution of SK channels, we propose that the elevated $IK_{Ca}$ in the *Shal-RNAi* background is primarily due to an increase in Slo-dependent $IK_{Ca}$ (see also below). We also observed a significant change in the delayed rectifier current ($IK_{DR}$) in MN1 expressing *Shal-RNAi* (*Figure 2A*). This effect parallels similar changes in $IK_{DR}$ in Kv4.2 knockout cardiac myocytes (*Guo et al., 2005*) and pyramidal neurons in mice (*Nerbonne et al., 2008*). The $IK_{DR}$ current can be encoded by four genes in *Drosophila* including: *Shab*, *Shaw*, *Shawl* and *KCNQ*. Pharmacological tools to dissect the function of each individual gene do not exist. However, the drug XE991 is a potent and selective inhibitor of KCNQ channels in both mammals and Drosophila (*Cavaliere and Hodge, 2011*; *Wang et al., 1998*). Application of XE991 (10 μM) diminished IKDR in wild type MN1, but there was no differential effect following *Shal-RNAi* (data not shown).

## The Krüppel transcription factor is essential for FRH following loss of Shal

We previously demonstrated that expression of the *Krüppel* (*Kr*) transcription factor is induced by genetic depletion or pharmacological inhibition of the Shal channel (*Parrish et al., 2014*). Kr expression is virtually absent in the wild type third instar CNS, but becomes highly expressed following loss of Shal (*Parrish et al., 2014*). Furthermore, over-expression of Kr in post-mitotic neurons is sufficient to drive changes in ion channel gene expression (*Parrish et al., 2014*). However, the role of Kr has never been studied at the level of somatic firing rates, nor has ion channel function been addressed. Therefore, it remains unknown whether Kr actually participates in the mechanisms of FRH. More specifically, it remains unclear to what extent Kr-dependent control of ion channel transcription influences the remodeling of ionic conductances during FRH. Indeed, we have previously documented that ion channel gene expression changes following loss of Shal (*Parrish et al., 2014*), but causal links to changes in ionic conductances have yet to be established. Finally, it remains unknown if the effects of Kr can be cell autonomous, or whether it acts through intercellular signaling intermediates.

If Kr is required for homeostatic plasticity, then loss of Kr in the *Shal* background should enhance firing rates, similar to what we observed with acute pharmacological block of $IK_A$ (*Figure 1D*). We quantified firing rates in MN1 in four conditions: 1) wild type, 2) *Kr-RNAi*, 3) *Shal-RNAi*, and 4) co-expression of *Shal-RNAi* and *Kr-RNAi*. Firing rates are equivalent when comparing wild type and *MN1-GAL4 > Shal* RNAi animals (*Figure 3—figure supplement 1A*). Firing rates are also unchanged when comparing wild type and *MN1-GAL4 > Kr* RNAi animals (*Figure 3—figure supplement 1B*). This is an important control, demonstrating that post mitotic knockdown of Kr, a master regulator of cell fate in the embryo, has no baseline effect. However, when Kr and Shal are simultaneously knocked down in MN1, firing rates were significantly decreased compared to wild type at all current steps above 25 pA (*Figure 3A & B*). These data are consistent with the conclusion that induction of Kr expression following loss of Shal is required for FRH. It was surprising, however, that firing rates were depressed compared to wild type, rather than enhanced, as predicted.

## Kr controls the homeostatic regulation of action potential waveform

In the *Shal-RNAi* condition, action potential (AP) waveforms are indistinguishable from wild type, arguing for preservation of AP waveform during FRH (*Figure 3C,D & E*). We found that *Kr-RNAi* has no effect on AP waveform. However, when Shal and Kr were simultaneously knocked down in MN1, AP waveforms were significantly altered (*Figure 3C & D*). Specifically, the after-hyperpolarization

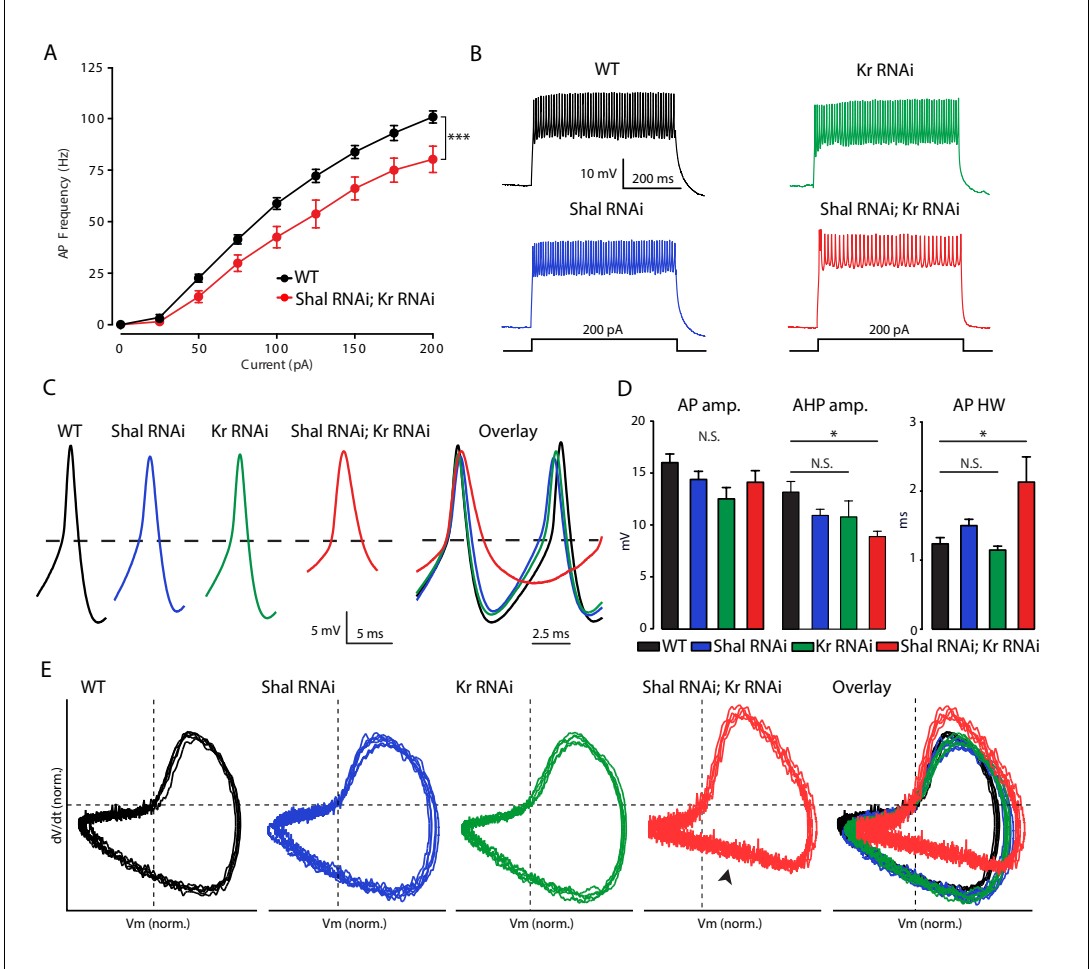

**Figure 3.** Krüppel is necessary for firing rate homeostasis and preservation of AP waveform. (**A**) F-I curves of WT (black; n = 10) and double *Shal-RNAi; Kr-RNAi* (red; n = 16). (**B**) Representative voltage traces from WT (black), *Shal-RNAi* (blue), *Kr-RNAi* (green) and double *Shal-RNAi; Kr-RNAi* (red) at 200 pA current injections. (**C**) Left: Example action potential waveforms for WT (black), *Shal-RNAi* (blue), *Kr-RNAi* (green) and double *Shal-RNAi; Kr-RNAi* (red). Right: Action potential overlays. (**D**) Action potential waveform quantification. (**E**) Phase plane plots of normalized example action potential waveforms for each genotype (left four panels) and overlays (far right panel). Each plot contains five sequential action potentials from a representative cell to illustrate AP to AP consistency. Mean ± S.E.M.; *p<0.05; **p<0.005; ***p<0.0005. N.S., not significant; two-way RM-ANOVA (F-I curves), one-way ANOVA with Bonferoni post-hoc tests (AP waveform comparisons).

DOI: https://doi.org/10.7554/eLife.45717.006

The following figure supplements are available for figure 3:

**Figure supplement 1.** Kr does not contribute to setting MN1 baseline firing rate.

DOI: https://doi.org/10.7554/eLife.45717.007

**Figure supplement 2.** Normal motoneuron morphology.

DOI: https://doi.org/10.7554/eLife.45717.008

amplitude was significantly smaller and AP half-width duration was significantly increased compared to wild type. These effects on AP waveform can be clearly observed when representative APs are overlaid (*Figure 3C*) and in phase-plane plots of representative action potentials (*Figure 3E*). The phase-plane plots were generated for five sequential APs from individual representative recordings from each genotype, selected as matching the average properties presented in *Figure 3D*. In particular, we note the reduced rate of repolarization in double *Shal-RNAi, Kr-RNAi* (*Figure 3E*, arrowhead). Thus, Kr participates in homeostatic control of both action potential waveform and firing rates following loss of Shal.

## MN1 cell identity is maintained following post-mitotic Kr knockdown

It is well established that Kr is a master regulator of cell fate determination in neurons (*Isshiki et al., 2001*) and other cell types (*McConnell and Yang, 2010*). But, the action of Kr in post-mitotic neurons is not understood. To confirm that we have not grossly altered cell fate, we examined MN1 morphology and passive-electric properties comparing *MN1-GAL4 > UAS* GFP (wild type) to the three genotypes analyzed throughout this paper: knockdown of Kr, knockdown of Shal, and simultaneous knockdown of both Kr and Shal. There was no change in MN1 cell number or gross morphology in the CNS (*Figure 3—figure supplement 2A,B,C & D*). We further measured somatic diameter and the width of the proximal dendrite as features that contribute to the passive electrical properties of these cells. No significant differences were observed (*Figure 3—figure supplement 2E & F*). Finally, we quantified cell capacitance and input resistance (*Figure 3—figure supplement 2G & H*). We found a decrease in input resistance for both *Kr-RNAi* and combined *Shal-RNAi, Kr-RNAi*. Although, the double *Shal-RNAi, Kr-RNAi* condition has an effect on input resistance, this cannot account for the difference in AP waveform or firing rates since *Kr-RNAi* alone matches wild type for both measures.

## Increased firing rate variance is associated impaired FRH

The observation that firing rates are decreased in the combined *Shal-RNAi, Kr-RNAi* condition could be due to MN1 acquiring a new firing rate set point or it could be due to the loss of homeostatic control. We reason that if a new set point is established, then the cell would target the new set point firing rate accurately, and the variance of firing rate would be equivalent to that observed in wild-type controls. By contrast, if FRH is disrupted by loss of Kr, then we expect to observe an increase in

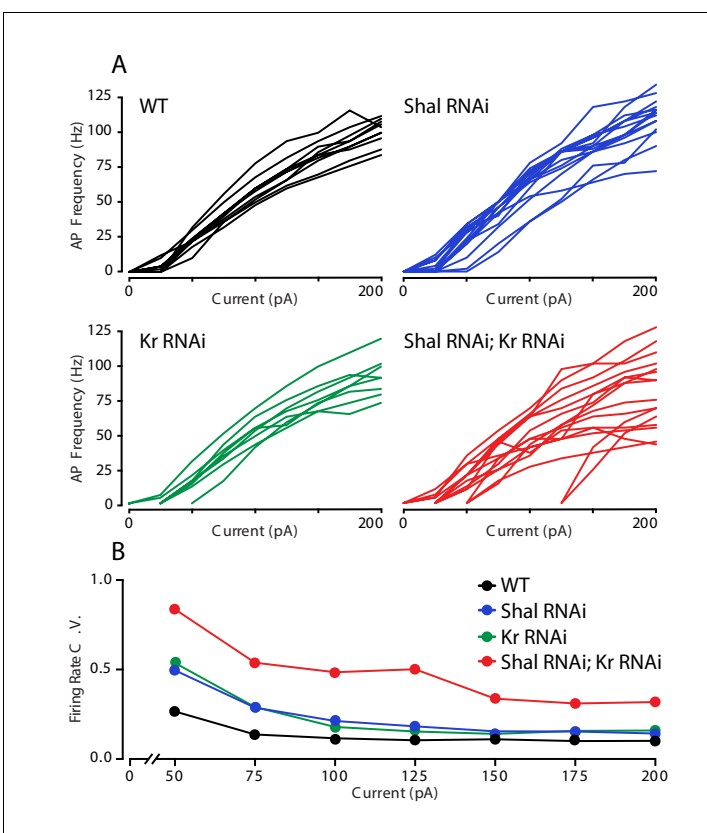

**Figure 4.** Krüppel constrains cell-to-cell firing rate variance. (**A**) Individual motorneuron F-I curves for WT (black), *Shal-RNAi* (blue), *Kr-RNAi* (green) and double *Shal-RNAi; Kr-RNAi* (red) groups. (**B**) The coefficient of variation across cells is calculated for each genotype at each current injection step and plotted. Data at 25 pA are not shown since a large fraction of cells in each genotype failed to fire an action potential.
DOI: https://doi.org/10.7554/eLife.45717.009

firing rate variability. We compared the F-I curves of individual MN1 neurons within each genotype (*Figure 4A*). It is clear that there was increased variability across cells in the double *Shal-RNAi, Kr-RNAi* condition compared to wild type and each individual knockdown alone. We quantified cell-to-cell variability across all current injections using the coefficient of variation (*Figure 4B*). The double *Shal-RNAi, Kr-RNAi* condition had the highest variability of all four genotypes, an effect that is not additive for current steps above 100 pA. These data are consistent with the hypothesis that Kr is essential for firing rate homeostasis, rather than revealing a new homeostatic set point. However, we acknowledge that the molecular basis for a homeostatic set point, in any system, has yet to be defined. Finally, it is worth noting that no individual cell ever fired at rates exceeding wild type as we observe following application of 4-AP (*Figure 1D*), indicating that the loss of firing rate homeostasis is not without some remaining constraint on firing frequencies in vivo.

## Kr selectively controls the homeostatic enhancement of IK$_{Ca}$

We next addressed the ionic conductances that are controlled by Kr. In principle, loss of Kr could specifically impair the homeostatic rebalancing of ion channel expression, or it could simply de-regulate gene expression and, thereby, non-specifically alter firing rates. We have shown that the two most prominent changes following loss of Shal are increases in IK$_{Ca}$ and IK$_{DR}$ (*Figure 2A & B*). Here, we demonstrate that the increase in IK$_{Ca}$ following loss of Shal was completely blocked by simultaneous Kr knockdown (*Figure 5C*). Importantly, Kr knockdown had no effect on baseline IK$_{Ca}$ (*Figure 5D*) or on voltage-gated calcium currents in the *Shal-RNAi* background (*Figure 5E*). Thus, Kr is required for the homeostatic enhancement of IK$_{Ca}$. To our knowledge, *Kr* is the first gene demonstrated to have a selective action for homeostatic changes in channel function, without altering baseline channel activity.

Next, we examined IK$_{DR}$. We found that IK$_{DR}$ remained elevated in the double *Shal-RNAi, Kr-RNAi* condition (*Figure 5A*), similar to that observed in *Shal-RNAi* alone (*Figure 2A*). This suggests that Kr does not control the homeostatic upregulation of IK$_{DR}$ following loss of Shal. We confirmed that Kr knockdown alone had no effect on baseline IK$_{DR}$ (*Figure 5B*). Finally, as another control, we quantified IK$_A$ in double *Shal-RNAi, Kr-RNAi* neurons and demonstrate that IK$_A$ was knocked down just as efficiently as when *Shal-RNAi* was driven alone (*Figure 5F*). Thus, any effect of the double RNAi is not a consequence of diluting GAL4-mediated expression of our transgenes. Taken together, our data argue that Kr has an activity that is required for the homeostatic rebalancing of IK$_{Ca}$, but not IK$_{DR}$. Thus, we conclude that loss of Kr participates in a specific facet of FRH induced by loss of Shal.

## The BK channel Slo is essential for maintenance of set point firing rates

We reasoned that if the decreased firing rate observed in double *Shal-RNAi, Kr-RNAi* neurons is due to a selective loss of IK$_{Ca}$, then acute pharmacological inhibition of IK$_{Ca}$ should also decrease firing rate. We bath applied the selective BK channel inhibitor paxilline (600 nM) to both wild-type and *Shal-RNAi* preparations, and observed significantly reduced firing rates in MN1 (*Figure 6A,B & C*). Paxilline reduced maximal firing rates by 34% on average, compared to the 12% reduction due to driving *Kr-RNAi* in the *Shal-RNAi* background. This difference in effect size is consistent with Kr specifically regulating the *increase* in IK$_{Ca}$ following loss of Shal rather than eliminating IK$_{Ca}$. The effects of paxilline on action potential waveform were also consistent with those seen in the double *Shal-RNAi, Kr-RNAi* condition. We observed decreased AHP amplitude and a larger AP half-width duration (*Figure 5D & E*). These data explain decreased firing rates in the double *Shal-RNAi, Kr-RNAi*. Loss of Slo-dependent AP repolarization leads to the observed broader action potential and shallower AHP, an effect that is predicted to impede recovery of sodium channels from inactivation and thus cause decreased firing rate.

## An alternate homeostatic mechanism is induced in pore blocked *Shal* mutants

We have shown that a pore-blocking, knock-in mutation (*Shal$^{W362F}$*) induces equally robust FRH when compared to elimination of Shal with expression of *Shal-RNAi*. Thus, we expected to observe identical changes in both IK$_{CA}$ and IK$_{DR}$. First, we demonstrate upregulation of IK$_{DR}$ in *Shal$^{W362F}$* (*Figure 7A*), consistent with this expectation. Furthermore, we found no significant change in IK$_{Ca}$

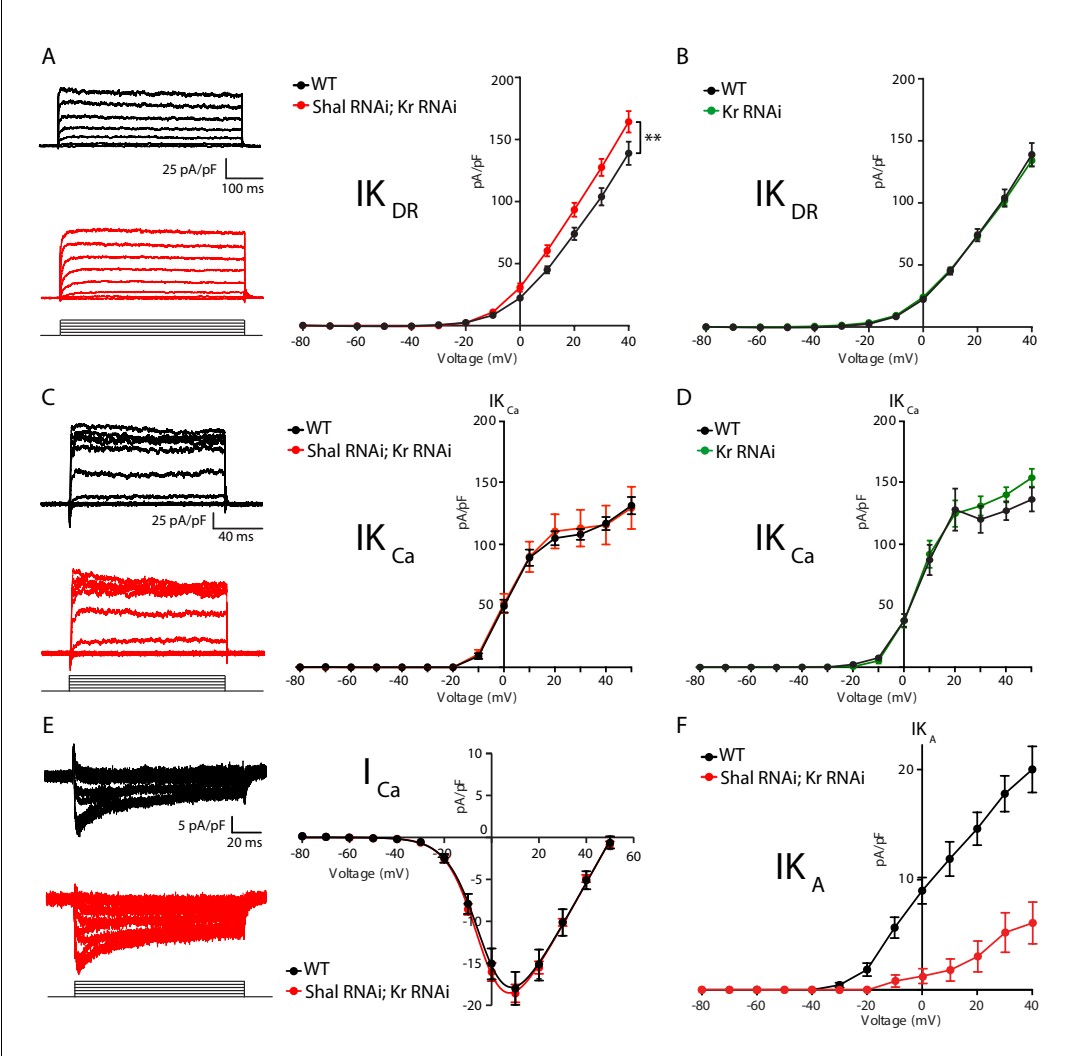

**Figure 5.** Krüppel controls IK$_{Ca}$ but not IK$_{DR}$ during firing rate homeostasis. (**A**) IK$_{DR}$ I-V plots (right) and representative traces (left) in WT (black, n = 7) and *Shal-RNAi; Kr-RNAi* (red, n = 19) MN1. (**B**) I-V plots of WT (black, n = 7) and *Kr-RNAi* (green, n = 14) MN1. (**C**) IK$_{Ca}$ I-V plots (right) and representative traces (left) in WT (black) and *Shal-RNAi; Kr-RNAi* (red) MN1. (**D**) I-V plots of WT (black, n = 7) and *Kr-RNAi* (green, n = 9) MN1. (**E**) I-V plots (right) and representative traces (left) of voltage dependent calcium currents (I$_{Ca}$) in WT (black; n = 8) and double *Shal-RNAi; Kr-RNAi* (red; n = 6) MN1. (**F**) I-V plots of MN1 IK$_A$ currents in WT (black, n = 20) and double *Shal-RNAi, Kr-RNAi* (red, n = 10). Mean ± S.E.M.; *p<0.05; **p<0.005; ***p<0.0005; two-way RM-ANOVA with Sidak post-hoc tests. Mean ± S.E.M.; *p<0.05; **p<0.005; ***p<0.0005. N.S., not significant; two-way RMANOVA (I-V plots), one-way ANOVA with Bonferoni post-hoc tests.

DOI: https://doi.org/10.7554/eLife.45717.010

(*Figure 7B*). We confirmed, via quantitative PCR, that *Slo* transcript is upregulated following loss of Shal protein (*Figure 7E*; see *Parrish et al., 2014* for initial observation). However, we did not observe a change in *Slo* transcript in the *Shal^{W362F}* non-conducting mutant (*Figure 7F*). In agreement, we observed a small but statistically significant broadening of the AP waveform in *Shal^{W362F}* (*Figure 7C & D*). Thus, FRH appears to be differentially achieved in *Shal^{W362F}* compared to *Shal-RNAi*.

One possibility is that the *Shal^{W362F}* pore blocking mutation induces a unique homeostatic solution. To assess this possibility, we used quantitative PCR to examine changes in gene expression for *Krüppel, Shaker, slo,* and *Shab*. The transcripts for all four of these genes are significantly elevated in the *shal* null mutant (*Shal^{495}; Parrish et al., 2014*). Here, we compare the *Shal* null mutant (*Shal^{495}*) to *Shal^{W362F}* since both directly alter the *Shal* gene locus and do so throughout the nervous system and throughout development. Confirming prior observations using gene expression arrays

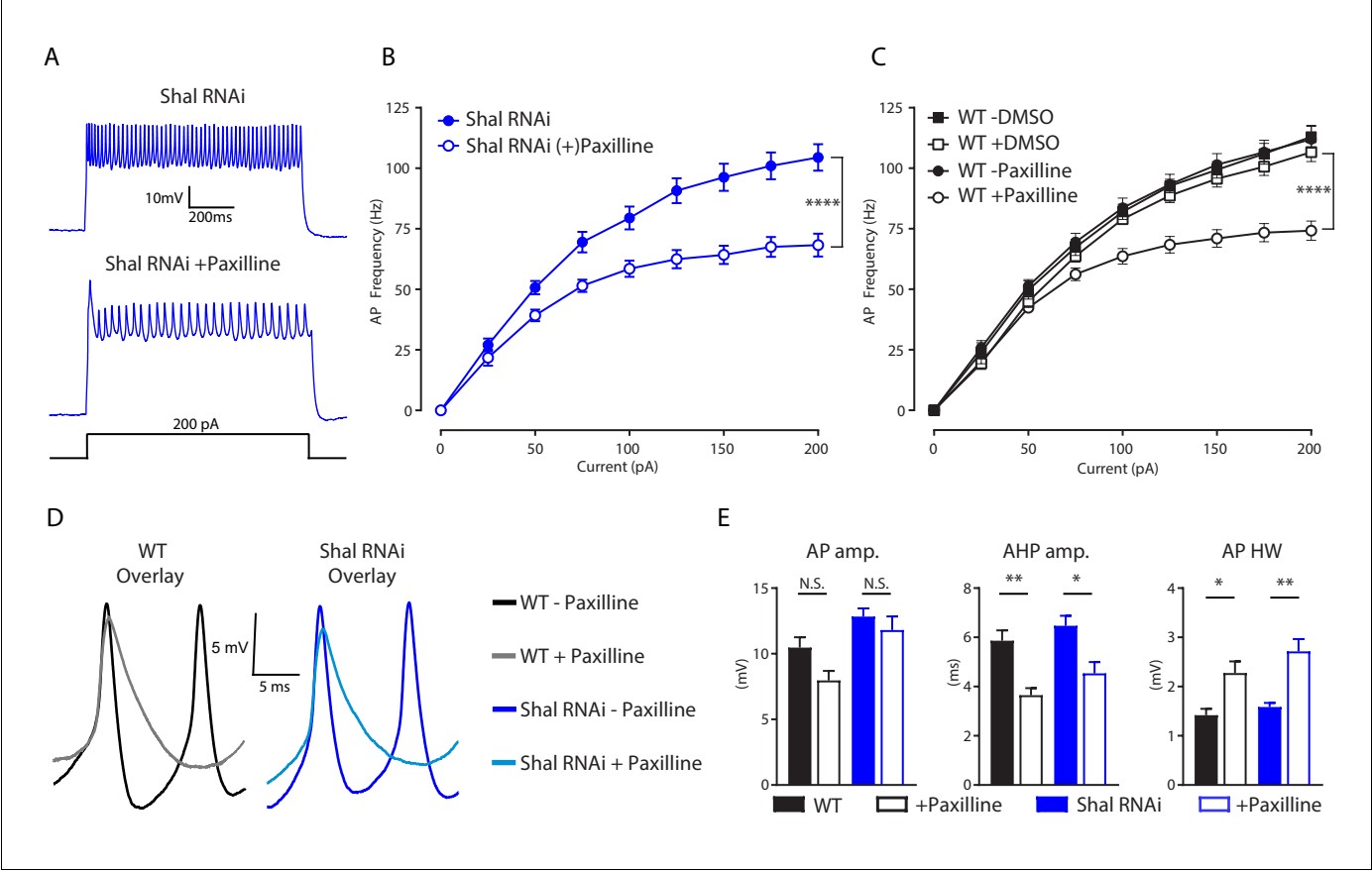

**Figure 6.** IK$_{Ca}$ is necessary to maintaining WT firing rates. (**A**) Representative voltage traces from *Shal-RNAi* and *Shal-RNAi* +Paxilline at 200 pA step current injection. (**B**) F-I curves of *Shal-RNAi* at baseline (filled circles) and *Shal-RNAi* +Paxilline (open circles), paired recordings, n = 8. (**C**) F-I curves of WT at baseline (filled circles) and WT + Paxilline (open circles), paired recordings, n = 10; WT at baseline (filled squares) and WT + DMSO (open squares), paired recordings, n = 6. Paxilline was dissolved in DMSO. (**D**) Example action potential waveforms for WT ± Paxilline overlayed (black, grey) and *Shal-RNAi* ± Paxilline overlayed (dark blue, light blue). (**E**) Action potential waveform quantification. Mean ± S.E.M.; *p≤0.05; **p<0.01; ***p<0.0001 ****p≤0.0001. N.S., not significant; two-way RM-ANOVA (F-I curves), one-way ANOVA (AP waveform comparisons) with Tukey post-hoc tests.

DOI: https://doi.org/10.7554/eLife.45717.011

(*Parrish et al., 2014*), the transcription all four genes was increased in the *Shal* null mutant (*Figure 7E*). However, none of these genes showed altered expression in *Shal^{W362F}* (*Figure 7F*, bottom). We then extended this analysis in *Shal^{W362F}* to include *KCNQ*, *Shaw* and *Shawl* (*Figure 7F*, top). Again, there was no change in the expression of these channels, whereas *KCNQ* was upregulated in the *Shal* null (*Parrish et al., 2014*). Thus, eliminating Shal using either a null mutation or via expression of *Shal-RNAi* initiates FRH that is achieved by induction of the Krüppel transcription factor followed by Krüppel-dependent enhancement of IK$_{Ca}$ current and Krüppel-independent enhancement of IK$_{DR}$ current, as well as increased transcription of several other ion channel genes. By contrast, in the *Shal^{W362F}* mutant, equally robust FRH is achieved by a selective increase in the IK$_{DR}$ current without a change in the expression of major IK$_{DR}$ genes. Thus, it appears that two separable, equally robust, homeostatic solutions are achieved downstream of different mutations in a single ion channel gene.

## Solution-specific effects on motor behavior

Wild-type, *Shal^{W362F}*, and Shal knockdown animals were individually tested for motor behavior in a negative geotaxis assay (*Figure 7G,H*). Negative geotaxis is a powerful innate behavior that can be used to assess adult *Drosophila* motility and coordination without confounding effect of motivation or learning. All wild-type and *Shal^{W362F}* flies climbed up the walls of a glass vial above 10 cm within 10 s. No statistically significant differences in average climbing speed were detected, consistent with

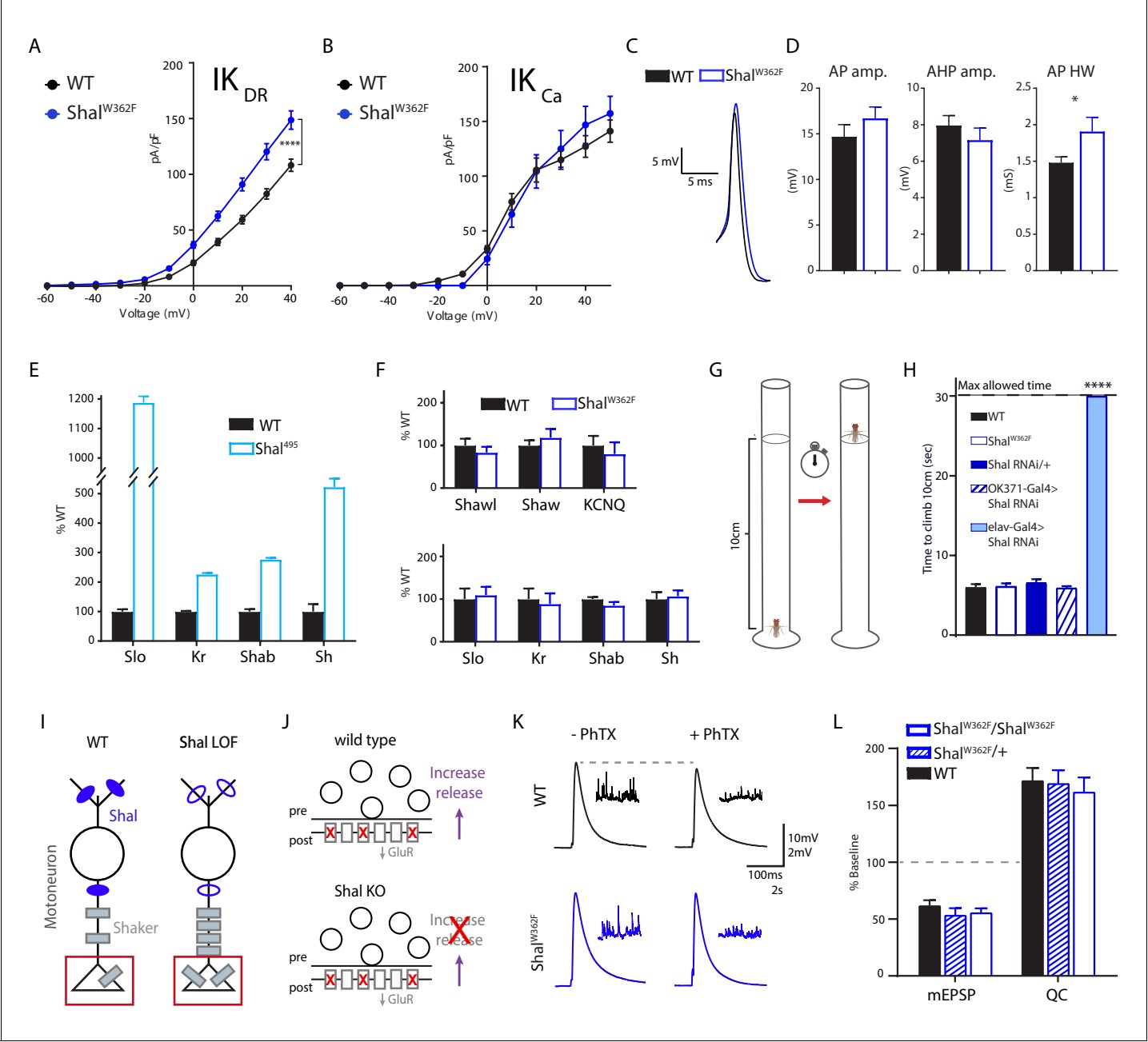

**Figure 7.** Shal activity block engages distinct homeostatic rebalancing mechanism. (**A**) IK_DR I-V plots (left) and representative traces (right) in WT (black, n = 7) and *Shal^W362F* (blue, n = 13) motoneurons. (**B**) IK_Ca I-V plots (left) and representative traces (right) in WT (black; n = 12) and *Shal^W362F* (blue; n = 10) motoneurons. (**C**) Example action potential waveforms for WT (black) and *Shal^495* (blue) overlaid. (**D**) Action potential waveform quantification. (**E**) Quantitative PCR from WT (black) and *Shal^495* (blue) whole third instar larval brains (≥3 biological replicates, each). (**F**) Quantitative PCR from WT (black) and *Shal^W362F* (blue) whole third instar larval brains (≥3 biological replicates, each). (**G**) Cartoon representation of negative geotaxis assay. A single animal was placed in a 20 cm tall clean glass tube. The fly was startled by tapping and time to climb to 10 cm high mark was recorded. (**H**) Results of climbing assay. WT n = 10, *Shal^W362F* n = 10, *Shal-RNAi/+* n = 10, *Shal-RNAi/OK371* n = 10, *elav-Gal4;Shal-RNAi* n = 10, *elav-Gal4;Shal-RNAi; Kr-RNAi* n = 10. (**I**) Cartoon diagramming how loss of Shal induces compensatory upregulation of Shaker in motoneuron axons and nerve terminals. (**J**) Diagrams illustrating the consequences of ion channel rebalancing on presynaptic homeostatic potentiation (PHP). Top: reduced postsynaptic receptor sensitivity triggers compensatory upregulation of presynaptic neurotransmitter release during PHP. Bottom: Shal knockout animals fail to express PHP. (**K**) Sample traces showing representative EPSP and mEPSP amplitudes in the absence and presence of PhTX for WT (black) and *Shal^W362F* (blue). (**L**) Reduction in mEPSP amplitudes and increase in quantal content (QC) following incubation in PhTX expressed as percent change relative to baseline for each genotype in absence of PhTX. WT baseline n = 9,+PhTX n = 12; *Shal^W362F/+*baseline n = 6,+PhTX n = 6; *Shal^W362F/ShalW^W362F* baseline

*Figure 7 continued on next page*

*Figure 7 continued*

n = 9,+PhTX n = 10. Mean ± S.E.M.; *p<0.05; **p<0.005; ***p<0.0005, ****p<0.0001. N.S., not significant; two-way RM-ANOVA with Bonferoni post-hoc tests (I-V plots); Student's t test (AP waveform comparisons); one-way ANOVA (climbing assay) with Tukey post-hoc tests.

DOI: https://doi.org/10.7554/eLife.45717.012

the idea that FRH enables normal motor behavior. Since the $Shal^{W362F}$ mutation is present throughout the CNS, throughout development, we initially compared $Shal^{W362F}$ to pan-neuronal Shal knockdown (*elav-Gal4; Shal-RNAi*), expecting similarly robust motor behavior. This was not observed. Instead, pan-neuronal Shal knockdown dramatically altered animal behavior, and every animal that was tested failed to ascend past the 10 cm mark within 30 s, the maximal allotted time. The *shal* null mutation behaves similarly, being unable to climb the walls of a vial (not shown). Control animals (*Shal-RNAi/+*) were no different from wild type. As a further control, we assessed motoneuron-specific *Shal* knockdown (*OK371-Gal4*). In this experiment, climbing behavior was wild type, again consistent with the conclusion that FRH, which we demonstrate in motoneurons, is sufficient to restore normal animal behavior. Why is motor behavior differentially affected in pan-neuronal Shal knockdown and *Shal* null mutants compared to the pan-neuronal effects of the $Shal^{W362F}$ mutation? One possibility is that every neuronal cell type is able to engage the form of homeostatic plasticity triggered by the $Shal^{W362F}$ mutation, but not every cell type engages FRH equivalently following deletion of the Shal protein. Alternatively, loss of Shal protein triggers a homeostatic response that includes changes in ion channel gene expression, and in some cell types the altered expression of ion channel genes could lead to maladaptive effects on circuit function and animal behavior (see discussion).

## Solution-specific effects on presynaptic homeostatic plasticity

We previously demonstrated that FRH, induced by pan-neuronal knockdown of Shal, interferes with the subsequent induction of presynaptic homeostatic plasticity (PHP), assayed at the neuromuscular junction (*Bergquist et al., 2010*). A current model for this interference effect is that increased Shaker expression, caused by the presence of the protein null $Shal^{495}$ mutation or by direct over-expression of a *Shaker* transgene, blocks the expression of PHP that is selectively induced at the NMJ (*Figure 7I & J*; *Bergquist et al., 2010*; *Parrish et al., 2014*). Increased levels of Shaker channel, which localizes to the presynaptic terminal at the NMJ (*Figure 7*), prevents ENaC channel-dependent depolarization of the presynaptic plasma membrane, which is necessary to increase presynaptic calcium influx that drives enhanced neurotransmitter release during PHP (*Younger et al., 2013*; *Parrish et al., 2014*; *Orr et al., 2017*). If this model is correct, then FRH induced by the $Shal^{W362F}$ mutation should have no effect of the rapid induction of PHP because the mechanisms of FRH in the $Shal^{W362F}$ mutation do not include a change in *Shaker* expression. This is precisely what we observe. We induced PHP by incubating the NMJ in a sub-blocking concentration of the glutamate receptor antagonist philanthotoxin-433 (PhTX; 15–20 μM) according to well-established protocols (*Frank et al., 2006*). Decreased mEPSP amplitude was precisely offset by an increase in presynaptic release (quantal content, QC) that restored EPSP amplitudes to pre-PhTx amplitudes in both wild type and $Shal^{W362F}$ animals (*Figure 7K & L*). Thus, unlike in the protein null $Shal^{495}$ mutant, synapses in the pore-blocking $Shal^{W362F}$ mutant were capable of undergoing PHP. Thus, the differential expression mechanisms of FRH induced by loss of Shal protein versus the $Shal^{W362F}$ mutation can have different effects on synaptic transmission and synaptic plasticity.

## Discussion

Here, we advance our mechanistic understanding of FRH in several ways. First, we demonstrate that FRH can be induced and fully expressed in single, genetically identified neurons. Since changes in the activity of a single motoneuron are unlikely to dramatically alter the behavior of the larvae, these data argue strongly for cell autonomous mechanisms that detect the presence of the ion channel perturbation and induce a corrective, homeostatic response. Second, we demonstrate that FRH functions to preserve the waveform of individual action potentials. This argues for remarkable precision in the homeostatic response. Third, we provide new evidence that the transcription factor

*Krüppel* is essential for FRH, and selectively controls the homeostatic enhancement of IK$_{CA}$, without altering the baseline ion channel current. Finally, we demonstrate that different mechanisms of FRH are induced depending upon how the *Shal* current is eliminated, and these differential expression mechanisms can have perturbation-specific effects on animal behavior.

We propose the existence of parallel homeostatic mechanisms, responsive to differential disruption of the *Shal* gene. We observe different compensatory responses depending upon whether the Shal protein is eliminated or the Shal conductance is eliminated. The following evidence supports the functional equivalence of our manipulations. First, the *Shal*$^{W362F}$ mutation completely eliminates somatically recorded IK$_A$ (*Figure 1*). Second, we demonstrate a dramatic reduction in IK$_A$ when *Shal-RNAi* is driven by *MN1-GAL4* in a single, identified neuron. Notably, the current-voltage relationship observed for *Shal-RNAi* is identical to that previously published for the *Shal*$^{495}$ protein null mutation, being of similar size and voltage trajectory including a + 50 mV shift in voltage activation (*Bergquist et al., 2010*). This remaining, voltage-shifted, IK$_A$-like conductance is attributed to the compensatory up-regulation of the Shaker channel on axonal membranes (*Bergquist et al., 2010*; *Parrish et al., 2014*) an effect that does not occur in the *Shal*$^{W362F}$ mutant (*Figure 7*). Thus, it seems reasonable to assume that Shal protein elimination and Shal conductance blockade initially create identical effects on neuronal excitability by eliminating Shal function. Subsequently, these perturbations trigger divergent compensatory responses. But, we acknowledge that we lack direct information about the immediate effects of the two perturbations.

## Comparison with prior studies of FRH in *Drosophila*

We define FRH as the restoration of neuronal firing rate in the continued presence of a perturbation. This definition is important because it necessitates that the underlying molecular mechanisms of FRH must have a quantitatively accurate ability to adjust ion channel conductances such that firing rate is precisely restored. Mechanistically, a prior example of FRH involves an evolutionarily conserved regulation of sodium channel translation by the translational repressor Pumillio (*Mee et al., 2004*; *Muraro et al., 2008*). This work, originally pursued in *Drosophila,* was extended to mouse central neurons where it was shown that Pumilio-dependent bi-directional changes in the sodium current occur in response to altered synaptic transmission, initiated by application of either NBQX or Gabazine (*Driscoll et al., 2013*). These data highlight the emerging diversity of molecular mechanisms that can be induced and participate in the execution of FRH (*Goold and Nicoll, 2010*; *Joseph and Turrigiano, 2017*).

It is necessary to compare our current results with prior genetic studies of the Shal channel in *Drosophila*. A prior report, examining the effects of partial *Shal* knockdown in larval motoneurons, observed a trend toward an increase in the sustained potassium current, but concluded no change (*Schaefer et al., 2010*). However, the small sample size for potassium current measurements in that study (n = 3 cells) and the incomplete *Shal* knockdown that was achieved, likely conspired to prevent documentation of the significant increase in IK$_{DR}$ that we observe (IK$_{CA}$ was not measured in *Schaefer et al., 2010*). A second prior study examined over-expression of a pore-blocked *Shal* transgene in cultured *Drosophila* embryonic neurons, revealing elevated firing rate and a broadened action potential (*Ping et al., 2011*). This was interpreted as evidence against the existence of FRH (*Ping et al., 2011*). However, neuronal precursors were cultured from 5 hr embryos (*Ping et al., 2011*), prior to establishment of neuronal cell fate and prior to the emergence of IK$_A$ currents in vivo, which occurs ~10 hr later in development (*Baines and Bate, 1998*). It remains unclear whether these cultured neurons are able to achieve a clear cell identity, which may be a prerequisite for the expression of homeostatic plasticity (*Davis, 2006*; *Davis, 2013*). Another possibility concerns the time-course of FRH, which remains uncertain. Finally, over-expression of the transgene itself might interfere with the mechanisms of FRH (*MacLean et al., 2003*), emphasizing the importance of the scarless, CRISPR-mediated gene knock-in approach that we have employed.

## Distinct homeostatic mechanisms downstream of a single ion channel gene

It is clear from studies in a diversity of systems that FRH can be induced by perturbations that directly alter neuronal activity without genetic or pharmacological disruption of ion channels or neurotransmitter receptors. For example, monocular deprivation induces an immediate depression of

neuronal activity in the visual cortex, followed by restoration of normal firing rates (*Hengen et al., 2013*). Research on the lobster stomatogastric system ranging from experiments in isolated cell culture (*Turrigiano et al., 1994*) to de-centralized ganglia (*Zhang et al., 2009*) have documented the existence of FRH that is consistent with an activity-dependent mechanism. It is equally clear that FRH can be induced by the deletion of an ion channel gene, including observations in systems as diverse as invertebrate and vertebrate central and peripheral neurons and muscle (*Swensen and Bean, 2005*; *Muraro et al., 2008*; *Andrásfalvy et al., 2008*; *Nerbonne et al., 2008*; *Van Wart and Matthews, 2006*; *Bergquist et al., 2010*; *Parrish et al., 2014*; *Driscoll et al., 2013*). But, it has remained unknown whether FRH that is induced by changes in neural activity is governed by the same signaling process that respond to ion channel gene mutations. Our current data speak to this gap in knowledge.

We demonstrate that changes in neural activity *cannot be solely* responsible for FRH. We compare two different conditions that each completely eliminate the Shal ion channel conductance and, therefore, are expected to have identical effects on neuronal excitability. We demonstrate robust FRH in both conditions. However, two separate mechanisms account for FRH. *Shal-RNAi* induces a transcription-dependent homeostatic signaling program. There is enhanced expression of *Krüppel* and a *Krüppel*-dependent increase in the expression of the *slo* channel gene and enhanced $IK_{CA}$ current. By contrast, the $Shal^{W362F}$ mutant does not induce a change in the expression of *Krüppel, slo* or any of five additional ion channel genes. Instead, we observe a change in the $IK_{DR}$ conductance, the origin of which we have yet to identify, but which appears to be independent of a change in ion channel gene transcription.

We propose the existence of two independent homeostatic signaling systems, induced by separate perturbations to the Shal channel gene. First, we propose that *Shal-RNAi* and the *Shal* null mutation trigger a homeostatic response that is sensitive to the absence of the Shal protein. In essence, this might represent an ion channel-specific system that achieves channel proteostasis, a system that might normally be invoked in response to errors in ion channel turnover (*Figure 8*). We speculate that many, if not all ion channels could have such proteostatic signaling systems in place. In support of this idea, the induction of *Kr* is specific to loss of Shal, not occurring in eight other ion channel mutant backgrounds, each of which is sufficient to alter neural activity, including *eag*, *para*, *Shaker*, *Shab*, *Shawl*, *slo*, *cac* and *hyperkinetic* (*Parrish et al., 2014*). Each of these channel mutations is well established to alter neuronal activity (*Srinivasan et al., 2012*; *Lin and Baines, 2015*; *Frolov et al., 2012*; *Kim et al., 2017*; *Kadas et al., 2015*; *Kawasaki et al., 2000*; *Stern and Ganetzky, 1989*). But, Kr responds only to loss of *Shal*.

Next, we propose that eliminating the Shal conductance in the $Shal^{W362F}$ mutant background induces a separable mechanism of FRH that is independent of ion channel transcription. While the mechanisms of this homeostatic response remain unknown, it is tempting to speculate that this mechanism is activity dependent, consistent with data from other systems cited above (*Figure 8*). Finally, it remains possible that these homeostatic signaling systems are somehow mechanistically linked (*Figure 8*). If so, this might provide a means to achieve the precision of FRH. For example, changes in ion channel gene expression might achieve a crude re-targeting of set point firing rates, followed by engagement of activity-dependent processes that fine tune the homeostatic response (*Figure 8*). Notably, distinct, interlinked negative feedback signaling has been documented in cell biological systems, suggesting a common motif in cell biological regulation (*Brandman and Meyer, 2008*).

An interesting prediction of our model is that activity-dependent mechanisms of FRH could be constrained by the action of the channel-specific homeostatic system. For example, loss of *Shal* induces a Shal-specific gene expression program and activity-dependent homeostatic signaling would be constrained to modulate the Shal-specific response. As such, the homeostatic outcome could be unique for mutations in each different ion channel gene. Given this complexity, it quickly becomes possible to understand experimental observations in non-isogenic animal populations where many different combinations of ion channels are observed to achieve similar firing rates in a given cell (*Marder and Prinz, 2002*; *Marder and Goaillard, 2006*; *O'Leary et al., 2014*; *Golowasch, 2014*). The combined influence of dedicated proteostatic and activity-dependent homeostatic signaling could achieve such complexity, but with an underlying signaling architecture that is different from current theories that focus on a single calcium and activity-dependent feedback processor.

**Figure 8.** Model for FRH in *Drosophila* motoneurons. A set point is operationally defined as the level of a variable that is retargeted by a homeostatic signaling system. We propose the existence of a set point for abundance of the Shal protein and a separate set point ▓▓▓▓▓▓▓▓▓ Each set point is connected to a dedicated sensor (S) that monitors either protein levels or ▓▓▓▓▓▓▓▓▓ sensor detects deviation from the set point, an error signal is produced. The sign and magnitude of the error is computed, over time, and drives changes in actuators (A) that implement negative feedback and restoration of the set point. Actuators in the Shal proteostasis feedback system (green) include the transcription factor *Kruppel* and downstream transcriptional changes in ion channels that include, but are not be limited to, *Shaker* and *slo.* Actuators for the activity-dependent homeostat (purple) remain unknown in *Drosophila*, but could include changes in ion channel transcription in other systems. We propose that the activity-dependent homeostat (purple) could be connected to the mechanisms of ion channel homeostasis. In this manner, activity-dependent homeostatic signaling could act primarily on the ion channel proteostatic program. If actuators of the activity-dependent proteostat include changes in ion channel gene expression, then channel proteostasis mechanisms could be secondarlily engaged. This might suggest the existence of repressors that couple channel and activity-dependent homeostatic systems, though there remains no experimental evidence to date.

DOI: https://doi.org/10.7554/eLife.45717.013

Finally, although we propose the existence of proteostatic feedback induced by the *Shal* null mutant and pan-neuronal RNAi, other possibilities certainly exist for activity-independent FRH, inclusive of mechanisms that are sensitive to channel mRNA (*MacLean et al., 2003*). For example, the transcriptional compensation that we document could be considered a more general form of 'genetic compensation' (*El-Brolosy and Stainier, 2017*). Yet, our data differ in one important respect, when compared to prior reports of genetic compensation. In most examples of genetic compensation, gene knockouts induce compensatory expression of a closely related gene. For example, it was observed that knockout of ß-actin triggers enhanced expression of other actin genes (*El-Brolosy and Stainier, 2017*) for review). The compensatory effects that we observe involve re-organization of the expression profiles for many, unrelated ion channel genes. Somehow, these divergent conductances are precisely adjusted to cover for the complete absence of the somato-dendritic A-type potassium conductance. Thus, we favor a more complex form of genetic compensation based upon homeostatic, negative feedback regulation (*Figure 8*).

## Kr-dependent control of IK$_{Ca}$

How does Kr-dependent control of IK$_{Ca}$ participate in FRH? IK$_{Ca}$ is a rapid, transient potassium current. Therefore, it makes intuitive sense that elevated IK$_{Ca}$ could simply substitute for the loss of the fast, transient IK$_A$ current mediated by Shal. If so, this might be considered an instance of simple genetic compensation (*El-Brolosy and Stainier, 2017*) for review). But, if this were the case, then blocking the homeostatic increase in IK$_{Ca}$ should lead to enhanced firing rates. This is not what we

observe. Instead, average firing rates decrease when Kr is eliminated in the background of *Shal-RNAi*. Thus, the Kr-dependent potentiation of $IK_{Ca}$ seems to function as a form of positive feedback, accelerating firing rate in order to achieve precise FRH, rather than simply substituting for the loss of Shal. Consistent with this possibility, acute pharmacological inhibition of $IK_{Ca}$ decreases, rather than increases, average firing rate. However, it should also be emphasized that the role of $IK_{Ca}$ channels in any neuron are quite complex, with context-specific effects that can either increase or decrease neuronal firing rates (*Contet et al., 2016*), for review). Indeed, it has been argued that BK channels can serve as dynamic range compressors, dampening the activity of hyperexcitable neurons and enhancing the firing of hypoexcitable neurons (*Contet et al., 2016*). This broader interpretation is also consistent with the observed Kr-dependent increase $IK_{Ca}$ during FRH.

In the stomatogastric nervous system of the crab, single-cell RT-PCR has documented positive correlations between channel mRNA levels, including transcript levels for IKCa and *Shal* (*Tobin et al., 2009*; *Temporal et al., 2014*; *Ransdell et al., 2012*). The molecular mechanisms responsible for the observed correlations remain unknown, but it seems possible that these correlations reflect a developmental program of channel co-regulation. Upon homeostatic challenge, the steady-state positive correlations are supplanted by homeostatic compensation, notably enhanced $IK_{Ca}$ in the presence of 4-AP. The pressing challenge is to define molecular mechanisms that cause the observed correlations and compensatory changes in ion channel expression during homeostatic plasticity. The Kr-dependent control of $IK_{Ca}$ following loss of *Shal* is one such mechanism. Clearly, there is additional complexity, as highlighted by the differential response to *Shal* null and *Shal* pore blocking mutations and the *pumilio*-dependent control of sodium channel translation in flies and mice (*Driscoll et al., 2013*; *Mee et al., 2004*; *Muraro et al., 2008*).

### The limits of FRH and implications for disease

Why do ion channel mutations frequently cause disease? If activity-dependent homeostatic signaling is the primary mechanism of FRH, then any ion channel mutation that alters channel function should be detected by changes in neural activity and firing rates restored. One possibility is that FRH is effective for correcting for an initial perturbation, but the persistent engagement of FRH might become deleterious over extended time. Alternatively, each solution could effectively correct firing rates, but have additional maladaptive consequences related to disease pathology. While this remains to be documented in disease, we show that loss of Shal protein throughout the CNS causes deficits in animal behavior that are not observed in animals harboring a pore-blocking channel mutation. Indeed, if one considers that FRH can include altered expression of a BK channel, the potential for maladaptive consequences is high. Altered BK channel function has been repeatedly linked to neurological disease including idiopathic generalized epilepsy (*Lorenz et al., 2007*), non-kinesigenic dyskinesia (*Du et al., 2005*) and Alzheimer's disease (*Beecham et al., 2009*; *Burns et al., 2011*; *Beecham et al., 2014*). Thus, there are potentially deleterious ramifications of altering BK channel expression if a homeostatic signaling process is engaged throughout the complex circuitry of the central nervous system. Although the phenotype of maladaptive compensation that we observe is clear, a block in synaptic homeostasis and impaired animal motility, there is much to be learned about the underlying cause. Ultimately, defining the rules that govern FRH could open new doors toward disease therapies that address these maladaptive effects of compensatory signaling.

## Materials and methods

**Key resources table**

| Reagent type (species) or resource | Designation | Source or reference | Identifiers | Additional information |
|---|---|---|---|---|
| Chemical compound, drug | Protease (Type XIV, *Streptomyces griseus*) | Sigma | P5147; CAS 9036-06-0 | |

*Continued on next page*

*Continued*

| Reagent type (species) or resource | Designation | Source or reference | Identifiers | Additional information |
|---|---|---|---|---|
| Chemical compound, drug | 1-naphthylacetyl spermine trihydrochloride (NASP) | Sigma | N193; CAS 1049731-36-3 | |
| Chemical compound, drug | Tetrodotoxin citrate (TTX) | Tocris | 1069; CAS 18660-81-6 | |
| Chemical compound, drug | Tetraethylamonium chloride (TEA-Cl) | Sigma | T2265; CAS 56-34-8 | |
| Chemical compound, drug/drug | 4-Aminopyridine (4-AP) | Sigma | A78403; CAS 504-24-5 | |
| Chemical compound, drug | Paxilline | Tocris | 2006; CAS 57186-25-1 | |
| Chemical compound, drug | XE-991 dihydrochloride | Tocris | 2000; CAS 122955-13-9 | |
| Chemical compound, drug | Philanthotoxin-433 (PhTX) | Santa Cruz Biotechnology | sc-255421; CAS 276684-27-6 | |
| Gene (*Drosophila melanogaster*) | $w^{1118}$ | N/A | FLYB: FBal0018186 | |
| Genetic reagent (*D. melanogaster*) | MN1-Ib-GAL4 | *Kim et al., 2009* | Yuh-Nung Jan (UCSF, San Francisco, CA) | |
| Genetic reagent (*D. melanogaster*) | UAS-Shal-RNAi | Vienna Drosophila RNAi Center (VDRC) | VDRC:103363 | P{KK100264}VIE-260B |
| Gene (*D. melanogaster*) | $Shal^{W362F}$ | This paper | N/A | CRISPR-Cas9 engineered point mutation |
| Genetic reagent (*D. melanogaster*) | $elav^{C155}$-GAL4 | Bloomington Drosophila Stock Center | BDSC:458 | P{w[+mW.hs]=GawB}elav[C155] |
| Genetic reagent (*D. melanogaster*) | OK371-GAL4 | Bloomington Drosophila Stock Center | BDSC:26160 | P{GawB}VGlut[OK371] |
| Gene (*D. melanogaster*) | $Slo^1$ | Bloomington Drosophila Stock Center | BDSC:4587 | |
| Genetic reagent (*D. melanogaster*) | UAS-Kr-RNAi | Bloomington Drosophila Stock Center | BDSC:27666 | P{TRiP.JF02745}attP2 |
| Genetic reagent (*D. melanogaster*) | UAS-CD8:GFP/UASm CD8:GFP | N/A | FLYB: FBti0012686 | |
| Gene (*D. melanogaster*) | $Shal^{495}$ | Bloomington Drosophila Stock Center | BDSC:18338 | PBac{WH}Shal[f00495] |

*Continued on next page*

*Continued*

| Reagent type (species) or resource | Designation | Source or reference | Identifiers | Additional information |
|---|---|---|---|---|
| Sequence-based reagent | Forward primer to clone *Shal* upstream gRNA into pCFD4 | This paper | N/A | TATATAGGA AAGATATCCGGG TGAACTTCGCAA CTTCACATCGAT TCCGGGTTTTAG AGCTAGAAATAGCAAG |
| Sequence-based reagent | Reverse primer to clone *Shal* downstream gRNA into pCFD4 | This paper | N/A | ATTTTAACTTGCT ATTTCTAGCTCTAA AACTCTGGCATTAG AGAACGATTCGACG TTAAATTGAAAATAGGTC |
| Sequence-based reagent | Forward primer for *Shal* 5' homology arm amplification, for insertion into pHD-Scarless DsRed | This paper | N/A | GGAGACCT ATAGTGTCTT CGGGGCCG Agcataattgctcccaagaac |
| Sequence-based reagent | Reverse primer for *Shal* 5' homology arm amplification, for insertion into pHD-Scarless DsRed | This paper | N/A | CGTCACAATATGATTATCT TTCTAGGGTTAACAAAA TGCACATACAAAAGATGC |
| Sequence-based reagent | Forward primer for *Shal* 3' homology arm amplification, for insertion into pHD-Scarless DsRed | This paper | N/A | CGCAGACTATCTTTC TAGGGTTAAGCGTT TTAGTTTTATCGAT TTATTTG |
| Sequence-based reagent | Reverse primer for *Shal* 3' homology arm amplification, for insertion into pHD-Scarless DsRed | This paper | N/A | GGAGACGTATAT GGTCTTCTTTTCC cgggaaacagccag ggggcgaggc |
| Sequence-based reagent | Primer for mutagenesis: *Shal* W362F and upstream PAM | This paper | N/A | CTTCACATCGATT CCGGCCGCCTTC TTTTATACCATC GTCACAATG |
| Sequence-based reagent | Primer for mutagenesis: downstream PAM | This paper | N/A | gttttttgttgatttca aatacactctggcat tagagaacg |
| Recombinant DNA reagent | pHD-Scarless DsRed | Drosophila Genomics Resource Center | DGRC:1364 | |
| Recombinant DNA reagent | pCFD4: U6:1-gRNA U6:3-gRNA | Addgene | 49411 | |
| Commercial assay or kit | RNeasy Plus Micro Kit | Qiagen | 74034 | |
| Commercial assay or kit | Turbo DNA-*free* Kit | Ambion | AM1907 | |
| Commercial assay or kit | SuperScript III First-Strand | Invitrogen | 18080–051 | |

*Continued on next page*

*Continued*

| Reagent type (species) or resource | Designation | Source or reference | Identifiers | Additional information |
|---|---|---|---|---|
| Commercial assay or kit | TaqMan Fast Universal PCR Master Mix (2X), no AmpErase UNG | Applied Biosystems | 4352042 | |
| Commercial assay or kit | KCNQ FAM Taqman gene expression assay | Applied Biosystems | Dm01846741_g1 | |
| Commercial assay or kit | Kr FAM Taqman gene expression assay | Applied Biosystems | Dm01821853_g1 | |
| Commercial assay or kit | RpL32 FAM Taqman gene expression assay | Applied Biosystems | Dm02151827_g1 | |
| Commercial assay or kit | Sh FAM Taqman gene expression assay | Applied Biosystems | Dm01828717_m1 | |
| Commercial assay or kit | Shab FAM Taqman gene expression assay | Applied Biosystems | Dm01821965_m1 | |
| Commercial assay or kit | Shaw FAM Taqman gene expression assay | Applied Biosystems | Dm01841512_g1 | |
| Commercial assay or kit | Shawl FAM Taqman gene expression assay | Applied Biosystems | Dm01809871_m1 | |
| Commercial assay or kit | Slo FAM Taqman gene expression assay | Applied Biosystems | Dm02150795_m1 | |
| Software, algorithm | Clampex 10.3 | Molecular Devices | https://www.moleculardevices.com | |
| Software, algorithm | Igor Pro 7.02 | WaveMetrics | https://www.wavemetrics.net/ | |
| Software, algorithm | MiniAnalysis 6.0.7 | Synapsoft | http://www.synaptosoft.com/MiniAnalysis/ | |
| Software, algorithm | SDS 2.4 | Applied Biosystems | https://www.thermofisher.com/order/catalog/product/4350490 | |
| Software, algorithm | Excel 2013 | Microsoft | https://www.microsoft.com/ | |
| Software, algorithm | GraphPad Prism 7 | GraphPad | https://www.graphpad.com/ | |
| Software, algorithm | Adobe Illustrator CC 2018 | ADOBE ILLUSTRATOR CC | https://www.adobe.com | |

## Fly stocks and genetics

In all experiments, the w1118 strain was used as the wild-type control. All fly stocks were maintained at 22–25°C and experimental fly crosses were raised at 25°C. w1118, *UASmCD8:GFP*, *OK371-GAL4*, *elav-GAL4*, *Slo[1]* and *Shal[495]* fly stocks were obtained from Bloomington Drosophila Stock Center.

The *Shal-RNAi* line (*KK100264*) was from the Vienna Drosophila RNAi Center (VDRC) and the *Kr-RNAi* line (JF02630) were from the Transgenic RNAi Project (TRiP) at Harvard Medical School. Motor neuron 1-specific RNAi gene knockdown was achieved by crossing the appropriate *UAS-RNAi* lines with a previously-reported *MN1-Ib-GAL4* driver line (*Kim et al., 2009*), gift from Yuh-Nung Jan. The *Shal*$^{W362F}$ mutant was engineered using the 'scarless' CRISPR-Cas9 gene editing method (*Gratz et al., 2015*), substituting phenylalanine for tryptophan at amino acid 362 at the endogenous *Shal* locus.

## Whole cell patch clamp electrophysiology

Whole-cell recordings were obtained from MN1-GFP motor neurons (*MN1-Ib-GAL4, UASmCD8: GFP*) in third-instar larvae. Larvae were prepared for electrophyiological recordings using standard larval fillet preps on a sylgard-coated recording chamber. External recording solution was perfused at 2–3 mL/min and contained (in mM): 135 NaCl, 5 KCl, 4 MgCl2, 5 HEPES, 1.5 CaCl2, pH 7.1, 295 mOsm. The glial sheath surrounding the ventral nerve cord was gently dissolved by local pipette application of 2% protease (Type XIV, *Streptomyces griseus*, Sigma) and the preparation was per-fused with recording solution for 10 min to wash away residual protease. 1-naphthylacetyl spermine trihydrochloride (NASP, 25 µM, Sigma) was washed on to the preparation to prevent muscle contrac-tion during the recording. Whole-cell recordings were obtained using standard thick-walled borosili-cate glass electrodes (4–6 MΩ, King Precision Glass) filled with appropriate internal solution for each experiment (see below). Whole cell patch clamp recordings were obtained with an Axon 700B (cur-rent clamp) or Axon 200B (voltage clamp) amplifiers (Molecular Devices), digitized at 20 kHz with a Digidata 1440A and recorded using Clampex 10.3. Recordings with series resistance greater than 15 MΩ and/or resting membrane potential more depolarized than −55 mV and/or input resistance less than 400 MΩ were discarded and excluded from analysis. All recordings were made at room temper-ature (20–22°C). All salts or other reagents used for electrophysiology were obtained from Sigma, unless noted otherwise.

## Current clamp

Whole cell patch clamp recordings were made using an intracellular solution containing (in mM): 140 Kmethanesulfonate, 5 KCl, 10 HEPES, 5 NaCl, 5 EGTA, 2 MgATP and 0.2 NaGTP, pH 7.35, 280–290 mOsm. Constant current was injected into cells to adjust Vm to between −50 and −55 mV. Cells requiring more than ±15 pA to set Vm were discarded from analysis. MN1 excitability was assessed by 500 ms square pulse current injections (−50 - + 200 pA, 25 pA/step). Frequency vs current (F-I) plots were constructed by calculating the firing rate for each current step and plotting verses current step amplitude.

## Voltage clamp

### General

All recorded currents were normalized to whole-cell capacitance, and current-voltage (I-V) plots were constructed by plotting measured current amplitudes verses respective voltage steps. The junction potential was measured for each internal solution and corrected in final I-V plots. Leak cur-rents were subtracted offline.

### Voltage Dependent Potassium Currents (IK)

IK currents were recorded with the same internal solution used for current clamp recordings. Tetro-dotoxin (TTX, 1 µM), CdCl2 (300 µM) were added to the external solution to block voltage-activated sodium and calcium channels, respectively. Cells were held at −70 mV after obtaining stable whole-cell configuration, and series resistance and capacitance were compensated (>85% predict./corr., 10 µS lag). A-type potassium currents (IK$_A$) were isolated by current subtraction following a two-phase voltage step protocol: 1) voltage steps from −90 to +40 mV (10 mV/step, 500 ms duration, 0.1 Hz inter-step interval), followed by 2) a 250 ms voltage pre-pulse to −30 mV to inactivate A-type potas-sium channels, followed by voltage steps from −90 to +40 mV (10 mV/step, 500 ms duration, 0.1 Hz inter-step interval). Delayed rectifier potassium currents (IK$_{DR}$) were measured with a 250 ms voltage pre-pulse to −30 mV to inactivate A-type potassium channels, followed by a voltage step protocol from −90 to +50 mV (10 mV/step, 500 ms duration, 0.1 Hz inter-step interval).

## Calcium-dependent potassium currents (IK$_{Ca}$)

IK$_{Ca}$ currents were recorded with the same internal solution used for current clamp recordings. Tetrodotoxin (1 µM) was added to the external solution to block voltage-activated sodium channels. K$_{Ca}$ currents were isolated by subtraction of the current traces recorded with a voltage step protocol from −90 to +50 mV (10 mV/step, 100 ms duration, 0.1 Hz inter-step interval) before and after CdCl2 (300 µM) application.

## Calcium currents (I$_{Ca}$)

External recording solutions were optimized for I$_{Ca}$ recordings, and contained (in mM): 100 NaCl, 5 KCl, 4 MgCl2, 30 tetraethylamonium chloride (TEA-Cl), 2 4aminopyridine (4-AP), 5 HEPES, 1.5 CaCl2, 1.5 BaCl2, 0.001 TTX; pH7.1, 295 mOsm. The intracellular solution contained (in mM): 125 Cs-methanesulfonate, 10 TEA-Cl, 5 4-AP, 10 HEPES, 4 NaCl, 5 EGTA, 2 MgATP, 0.2 MgGTP; pH 7.35, 285 mOsm. I$_{Ca}$ currents were recorded using a pre-pulse to −90 mV (1 s) followed by voltage steps from −90 to +50 mV (10 mV/step, 120 ms duration, 0.1 Hz inter-step interval). Ca2+ (1.5 mM, CaCl2) and Ba2+ (1.5 mM, BaCl2) were used as charge carriers to enhance macroscopic currents.

## Persistent sodium currents (I$_{NaP}$)

I$_{NaP}$ was measured according to previously described protocols (*Lin et al., 2009*; *Mee et al., 2004*). The external recording solution, optimized for I$_{Na}$ recordings, contained (in mM): 100 NaCl, 5 KCl, 50 TEA-Cl, 10 4-AP, 10 HEPES, 10 glucose, 0.5 CaCl2, 0.3 CdCl2 and 0.001 TTX; pH 7.1, 295 mOsm. The intracellular solution contained (in mM): 125 Cs-methanesulfonate, 10 TEA-Cl, 5 4-AP, 10 HEPES, 4 NaCl, 5 EGTA, 2 MgATP, 0.2 MgGTP; pH 7.35, 285 mOsm. I$_{NaP}$ currents were isolated with a pre-pulse voltage step protocol containing a conditioning step to +50 mV (50 ms) to inactivate fast transient I$_{Na}$ current spikes, followed by voltage steps from −70 to +50 mV (5 mV/step, 50 ms duration, 0.1 Hz inter-step interval) I$_{NaP}$. Persistent sodium currents were measured as the steady-state current at the end of each voltage step.

## Muscle recordings

Sharp electrode recordings were made from muscle six in abdominal segments two and three in third instar wandering larvae with an Axoclamp 900A amplifier (Molecular Devices), as described previously (*Frank et al., 2006*; *Müller et al., 2012*). Recordings were collected in HL3 saline containing (in mM): NaCl (70), KCl (5), MgCl2 (10), NaHCO3 (10), sucrose (115), trehalose (5), HEPES (5), and CaCl2 (0.3). Philanthotoxin-433 (PhTX; Sigma-Aldrich) was prepared as a stock solution (4 mM in DMSO) and diluted in HL3 saline to 16.6 µM. Semi-intact preparations with the CNS fat, and gut left intact were incubated in PhTX for ten minutes (*Frank et al., 2006*). Following the incubation, the larval preparations were rinsed and the dissection was completed as previously described (*Frank et al., 2006*). The motoneuron cut axon was stimulated as previously described (*Frank et al., 2006*). Cells depolarized more than −60 mV were excluded from analysis. Quantal content was calculated by dividing mean EPSP by mean mEPSP.

## Data analysis

Data were analyzed using custom procedures written in Igor Pro (Wavemetrics) and MiniAnalysis 6.0.0.7 (Synaptosoft). Statistical analysis was performed in Prism ($\alpha = 0.05$) and statistical tests used for each data set are indicated in figure legends.

## Quantitative RT-PCR

Primer probes for real-time PCR detection of *Kr*, *slo*, *Shab*, *Shaker*, *KCNQ*, *Shaw*, *Shawl*, and Ribosomal protein L32 (RpL32) were designed and developed by Applied Biosystems. RNA was isolated from the CNS of ≥15 third instar larvae per genotype (five brains for each of at least three biological replicates) using RNeasy Plus Micro Kit (Qiagen). A DNase digestion was performed to remove potential DNA contamination (TURBO DNA-free, Ambion). RNA was reverse-transcribed into cDNA (SuperScript III First-Strand synthesis system, Invitrogen). A no reverse transcriptase (RT) control was included for each sample. Purified cDNA was used as a template in PCR reaction with three 10 µl technical replicates for each condition (TaqMan Fast Universal PCR Master Mix, no AmpErase UNG, Applied Biosystems). Additionally, a 10 µl no RT reaction was included for each sample. The Applied

Biosystems 7900HT Fast Real-Time PCR System was used for all PCRs. Cycle Threshold (CT) was determined by automated threshold analysis using SDS2.4 software (Applied Biosystems, Foster City, CA). Relative gene expression levels between WT and mutant animals was determined using the ΔΔCT method. In brief, ΔCT values for experimental animals were subtracted from WT ΔCT values to obtain the ΔΔCT. Using the equation 2(-ΔΔCT)x100, the percent expression of each gene in the experimental condition relative to the control condition was calculated. Each experimental sample was compared to each wild-type sample.

## Negative geotaxis assay

All animals were raised at 25C. Animals were collected within 24 hr of eclosion and singly housed. On day 4 of life, animals were transferred to a glass cylinder with a marking 10 cm from the bottom. Animals were tapped to the bottom of the cylinder and the time to climb to the 10 cm marking was recorded. Three trials were performed for each animal and these times were averaged.

## Acknowledgements

Supported by NIH grant number R35NS097212 to GWD. We thank Davis lab members for assistance with data interpretation and analysis.

## Additional information

### Competing interests

Graeme W Davis: Reviewing editor, *eLife*. The other authors declare that no competing interests exist.

### Funding

| Funder | Grant reference number | Author |
| --- | --- | --- |
| National Institute of Neurological Disorders and Stroke | R35NS097212 | Graeme W Davis |

The funders had no role in study design, data collection and interpretation, or the decision to submit the work for publication.

### Author contributions

Yelena Kulik, Conceptualization, Data curation, Formal analysis, Validation, Investigation, Visualization, Methodology, Writing—original draft, Writing—review and editing; Ryan Jones, Conceptualization, Data curation, Formal analysis, Investigation, Visualization, Methodology, Writing—original draft, Writing—review and editing; Armen J Moughamian, Formal analysis, Investigation, Methodology, Writing—review and editing; Jenna Whippen, Data curation, Formal analysis, Methodology; Graeme W Davis, Conceptualization, Supervision, Funding acquisition, Methodology, Writing—original draft, Project administration, Writing—review and editing

### Author ORCIDs

Graeme W Davis http://orcid.org/0000-0003-1355-8401

### Decision letter and Author response

Decision letter https://doi.org/10.7554/eLife.45717.016
Author response https://doi.org/10.7554/eLife.45717.017

## Additional files

### Supplementary files

• Transparent reporting form
DOI: https://doi.org/10.7554/eLife.45717.014

### Data availability

All data generated or analysed during this study are included in the manuscript and supporting files.

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
