## [Decision Letter]

Thank you for submitting your article "Dual separable feedback systems govern firing rate homeostasis" for consideration by *eLife*. Your article has been reviewed by Ronald Calabrese as the Senior Editor, a Reviewing Editor, and three reviewers. The following individuals involved in review of your submission have agreed to reveal their identity: Ronald M Harris-Warrick (Reviewer #2); David J Schulz (Reviewer #3).

The reviewers have discussed the reviews with one another and the Reviewing Editor has drafted this decision to help you prepare a revised submission.

Summary:

In this manuscript the authors compare two different conditions that each reduce/eliminate the Shal ion channel conductance. They demonstrate robust firing rate homeostasis (FRH) in both conditions. However, two separate mechanisms account for FRH. *Shal-RNAi* induces a transcription-dependent homeostatic signaling program. There is enhanced expression of Kruppel and a Kruppel-dependent increase in the expression of the *slo* channel gene and enhanced IKCA current. By contrast, the Shal^W362F^ mutant does not induce a change in the expression of Krü*ppel, slo* or any of five additional ion channel genes. Instead, they observe a change in the IKDR conductance, the origin of which they have yet to identify, but which appears to be independent of a change in ion channel gene transcription. This is an important finding that sheds mechanistic light on observations in normal and pathological excitability showing that different homeostatic paths mediated homeostatic responses after different or even similar perturbations of the control (normal state).

Essential revisions:

There are several concerns about the interpretation of the results that require careful revision and clarification of the text and potentially figures but no new additional experiments.

1) Reviewer 1 (concern #1) points out that in Figure 1B there is remaining IK_A_ current at potentials at and above 0-10 mV for the Shal RNAi, but there is not remaining IK_A_ current in 1H for the Shal^W362F^. So, it seems that these two manipulations are not identical in terms of the remaining IK_A_ current (activity). Therefore, there is still some chance that the two different paths for achieving FRH might be due to one sensor detecting small differences in the amount of neural activity – and not to two different sensors detecting changes in protein or changes in activity.

In reviewer consultation it is was pointed out that the authors suggest that the residual current may result from distant axonal shaker currents. In Figure 7E, they show data from Shal^495^, a different null mutant that also results in no protein; there is a very large increase in shaker RNA, which is not seen in the Shal point mutation. So perhaps the difference is an upregulation of shaker current (which may or may not be properly targeted to axonal regions). If so, then indeed there may be differences in activity and responses to synaptic inputs in the two cases.

Conclusions, Introduction, Abstract, and particularly Discussion section should all be changed to accommodate these critiques. The Title should also be reconsidered.

2) Reviewer 2 (Discussion section concern #4) points out that the discussion does not make adequately clear the primary difference that triggers the two FRH mechanisms, which is the presence or absence of Shal protein.

3) Reviewer 3 (concern #1) points out that all of the work in this study (and many related) could be realistically classified as developmental plasticity, whereas much of the supporting literature cited evokes post-developmental plasticity and compensation in mature circuits. This distinction should be clarified and discussed.

---

## [Author Response]

Essential revisions:There are several concerns about the interpretation of the results that require careful revision and clarification of the text and potentially figures but no new additional experiments.1) Reviewer 1 (concern #1) points out that in Figure 1B there is remaining IK_A_ current at potentials at and above 0-10 mV for the Shal RNAi, but there is not remaining IK_A_ current in 1H for the Shal^W362F^. So, it seems that these two manipulations are not identical in terms of the remaining IK_A_ current (activity). Therefore, there is still some chance that the two different paths for achieving FRH might be due to one sensor detecting small differences in the amount of neural activity – and not to two different sensors detecting changes in protein or changes in activity.In reviewer consultation it is was pointed out that the authors suggest that the residual current may result from distant axonal shaker currents. In Figure 7E, they show data from Shal^495^, a different null mutant that also results in no protein; there is a very large increase in shaker RNA, which is not seen in the Shal point mutation. So perhaps the difference is an upregulation of shaker current (which may or may not be properly targeted to axonal regions). If so, then indeed there may be differences in activity and responses to synaptic inputs in the two cases.Conclusions, Introduction, Abstract, and particularly Discussion section should all be changed to accommodate these critiques. The Title should also be reconsidered.

We thank the reviewers for their feedback on this topic. Our response is several-fold.

First and foremost, the remaining IA-like current in Figure 1B reflects a homeostatic compensatory response to the absence of Shal protein. It does not reflect a difference in the underlying perturbation, which seems to be the concern raised by the reviewers. To be clear, the loss of Shal protein induces a compensatory, transcription-dependent program (Bergquist et al., 2010; Parrish et al., 2014). This program includes the increased expression of the Shaker channel. Importantly, the transcriptional and functional up-regulation of Shaker was demonstrated to be under the control of the Kruppel transcription factor (Parrish et al., 2014). The sequence of events is as follows: (1) Loss of Shal protein, (2) induction of Kruppel and (3) Kruppel-dependent increase in Shaker transcript. Note, that we previously provided functional evidence that Kruppel is necessary for an increase in Shaker function at the presynaptic terminal (Parrish et al., 2014). This information is also highlighted in our existing Figure 7, diagrammatically.

We previously provided evidence that the remaining IA-like current observed in the Shal protein null mutation reflects axonal Shaker (Bergquist et al., 2010: Parrish et al., 2014). Here is a quick summary: First, there are only two *Drosophila* genes that encode an IA current, Shal and Shaker.

We previously demonstrated that the Shal^495^ mutant is a protein null (Bergquist et al., 2010). So, in a Shal^495^ mutant, the only possible source of the remaining IA-like current is Shaker. Second, it is well established that Shaker is localized to the axon and presynaptic terminal in larval motoneurons of *Drosophila*, first documented by Lily Jan and later confirmed by our own work. Third, the remaining IA-like current is activated at +20mV, representing a 50mV depolarizing shift in voltage activation of an IA current. Thus, the IA-like current that remains in the Shal^495^ mutant must reflect the presence of axonal Shaker.

The new data in our paper further supports our conclusion that the axonal Shaker exists due to the compensatory, homeostatic transcriptional up-regulation of Shaker. Specifically, the poreblocked mutation lacks transcriptional up-regulation of Shaker transcript and also lacks the presence of the axonal, IA-like current.

So, based upon all of this accumulated evidence, from two prior papers and the data in our existing paper, we conclude that the two perturbations, loss of Shal protein and the Shal pore blocked mutant, are functionally equivalent because they both eliminate the Shal current. As highlighted by reviewer #2, the difference between these perturbations is that one eliminated the Shal protein and the other eliminates the Shal current. Elimination of the Shal protein, either in the Shal protein null (Shal^495^) or by RNAi (this paper; see also Parish et al., 2014) causes a Kruppel-dependent transcriptional, homeostatic program. The elimination of the Shal current does not induce the same Kruppel-dependent transcriptional program.

Finally, once the different homeostatic programs are initiated, in the different mutant backgrounds, there will be obvious differences in how the cells fully execute a homeostatic response. This fact is incorporated into our model. In our model, we connect the two homeostatic responses to illustrate this point.

We believe that we have been appropriately careful in our interpretation and conclusions. For example, in our abstract we state, “…we demonstrate robust FHR following either elimination of Kv4/Shal protein or elimination of the Kv4/Shal conductance.” This remains correct. Our title highlights ‘Dual and Separable’ feedback responses. Again, this remains correct. One response is transcriptional and requires the induction of the transcription factor Kruppel. The other does not. Thus, there are two, genetically separable effects. It is challenging, however, to introduce the entire argument (above) prior to the Discussion section of our paper. For example, this reviewer also would like us to cut back on introductory text in the Results section.

Furthermore, our transcriptional analysis of the pore-blocked mutation logically appears in Figure 7. We acknowledge, however, that further explanation is necessary given that this was a topic of reviewer conversation. Therefore, we have modified the text in two places, inclusive of the Results section and Discussion section:

1) We have added the following text to our Results section to provide further explanation regarding this issue: “By contrast, no substantial current was present in MN1 expressing *Shal-RNAi* until +20 mV, and voltage steps above +20 mV revealed only a small outward current with IK_A_ characteristics. Importantly, prior characterization of a Shal protein null mutation demonstrated the same current-voltage trajectory, including the same observed +50mV shift in voltage activation (Bergquist et al., 2010). In that prior study, the remaining, voltage-shifted, outward current was determined to reflect the homeostatic up-regulation of the Shaker channel, which resides in the electrotonically distant axonal membranes. This conclusion was independently confirmed in an additional, prior study (Parrish et al., 2014). Given these data, we conclude that *Shal-RNAi* effectively eliminated the relevant somatic IK_A_ that would participate in action potential repolarization.”

2) We have added the following text to our Discussion section. This paragraph also includes a statement, requested by reviewer #2, that highlights the nature of each perturbation: “We propose the existence of parallel homeostatic mechanisms, responsive to differential disruption of the Shal gene. We observe different compensatory responses depending upon whether the Shal protein is eliminated or the Shal conductance is eliminated. The following evidence supports the functional equivalence of our manipulations. First, the Shal^W362F^ mutation completely eliminates somatically recorded IK_A_ (Figure 1). Second, we demonstrate a dramatic reduction in IK_A_ when *Shal-RNAi* is driven by MN1-GAL4 in a single, identified neuron. Notably, the current-voltage relationship observed for *Shal-RNAi* is identical to that previously published for the Shal^495^ protein null mutation, being of similar size and voltage trajectory including a +50mV shift in voltage activation (Bergquist et al., 2010). This remaining, voltage-shifted, IK_A_-like conductance is attributed to the compensatory up-regulation of the Shaker channel on axonal membranes (Bergquist et al., 2010; Parrish et al., 2014) an effect that does not occur in the Shal^W362F^ mutant (Figure 7). Thus, it seems reasonable to assume that Shal protein elimination and Shal conductance blockade initially create identical effects on neuronal excitability by eliminating Shal function. Subsequently, these perturbations trigger divergent compensatory responses. But, we acknowledge that we lack direct information about the immediate effects of the two perturbations.”

2) Reviewer 2 (Discussion section concern #4) points out that the discussion does not make adequately clear the primary difference that triggers the two FRH mechanisms, which is the presence or absence of shal protein.

We have further emphasized this point through the addition of a new paragraph at the beginning of our Discussion section in which we state, “We observe different homeostatic responses depending upon whether the Shal protein is eliminated or the Shal conductance is eliminated.”

3) Reviewer 3 (concern #1) points out that all of the work in this study (and many related) could be realistically classified as developmental plasticity, whereas much of the supporting literature cited evokes post-developmental plasticity and compensation in mature circuits. This distinction should be clarified and discussed.

A) We have read the detailed comments of reviewer #3, Dr. Shultz. First, we would like to thank this reviewer for the generally positive statements and for making critical suggestions of our paper. Notably, in the detailed comments, Dr. Shultz points out that we did not reference an important study from his laboratory that is directly relevant to our current paper and we would like to apologize for this omission. The study by Ransdell, using pharmacology to acutely perturb ion channels in the stomatogastric ganglion to study homeostatic compensation, is nice work. We now cite this paper.

B) The reviewer asks for clarification of time course of homeostatic plasticity. We have added a sentence at the beginning of our discussion to acknowledge that we lack information about the developmental time course of the homeostatic response. However, we have no reason to suspect that there exist different homeostatic mechanisms depending upon the time of development, as suggested by this comment.